# Study on deformation law of coal pore mechanism characteristics under peak cluster landform

Jin Wang[1], Lulin Zheng[1,2]*, Hong Lan[2,3], Wenjibin Sun[1], Zhonglin Chen[3], Bo Li[3], Yujun Zuo[1], Yiping Zhang[1], Yi Sun[1], Fangbo Wen[1]

1 College of Minging, Guizhou University, Guiyang, China, 2 College of Resources and Environmental Engineering, Guizhou University, Guiyang, China, 3 Longfeng Coal Mine of Guizhou Lindong Coal Development Co., Ltd., Bijie, China

* llzheng@gzu.edu.cn

## Abstract

In order to reveal the change rule of coal pore structure under the peak cluster landform, coal samples were taken from nine different mountain heights based on the vertical variability of the landform, and the pore structure of the coal samples was tested using a combination of high-pressure mercuric pressure method and low-temperature nitrogen adsorption experiments. The results show that compared with the traditional coal reservoir, the pore structure of coal under the peak cluster landform, such as pore content, specific surface area and pore volume, changes with the change of vertical principal stress in a multi-peak state. The variations in the maximum and minimum values of vertical principal stress at each peak level are 1.04, 1.04, and 1.05 times, respectively. In terms of the adsorption pore volume, the differences between the maximum and minimum values are 2.30, 1.60, and 1.53 times, respectively. Notably, the degree of change between the peaks decreases as the peaks progress. Furthermore, with the increase in vertical principal stress, the degree of change in the specific surface area and pore volume of the corresponding adsorption pore between peaks also diminishes. It shows that the role of peak cluster landform conditions on coal pore structure is significant, and the extent of the role decreases with the increase of vertical principal stresses. Additionally, the vertical principal stress predominantly influences the fractal dimension $D_1$, which is represented as pore surface roughness. The capacity of coal samples for gas adsorption is mainly influenced by the roughness of the pore surfaces and the volume of the adsorption pores. In summary, the degree of microcrack formation in the pores of coal samples is influenced to some extent by the vertical elevation difference characteristics of the peak cluster landform, which not only controls the characteristics of the pore structure, but also affects the gas adsorption capacity of the coal samples. These results highlight the influence of vertical principal stress on coal pore closure and structural changes under the peak cluster landform. The results of the study can provide a reference for further research on further gas storage and enrichment laws, and the mine can judge the risk of protrusion for the gas accumulation capacity of

**Data availability statement:** All relevant data are within the manuscript and its Supporting Information files.

**Funding:** This research was supported by the Guizhou Province of Social Funding Project(LDLFJSFW2024-9). The funding had important roles in the study design, data collection and analysis. There was no additional external funding received for this study.

**Competing interests:** The authors declare that they have no known competing financial interests or personal relationships that could have appeared to influence the work reported in this paper.

coal under the peaked cluster landform, so as to formulate effective gas prevention and control measures in advance.

## 1. Introduction

Coal has long been a key player in China's energy sector, both in production and consumption [1,2]. With the ongoing rise in coal production, the depth of coal mining has also been increasing, resulting in higher levels of ground stress, gas pressure, and gas content in mines. This escalation heightens the risk of gas accidents in coal mining operations [3]. Practical experiences have demonstrated that the strategic and efficient extraction of coal seam gas is essential in mitigating gas accidents and maximizing gas resource recovery [4,5].

Coal and gas outbursts are considered a dynamic phenomenon that involves a complex kinetic evolution process. This process includes energy accumulation, transfer, irregular release, and structural damage caused by continuous superimposed forces [6–8]. The process behind coal and gas outbursts remains under investigation; however, the prevailing theory is known as the comprehensive action hypothesis. This theory proposes that outbursts are caused by the interaction of geological pressure, the presence of gases, the physical and mechanical characteristics of coal, the microscopic structure of coal, and gravitational influences. [9]. The stresses that influence coal and gas outbursts can be categorized into self-weight stress of overlying strata, tectonic stress, and dynamic stresses induced by mining activities [10,11]. Geological stress is a key factor in gas pressure and coal and gas outbursts, while tectonic stress significantly impacts the occurrence and migration of gas in these events [12,13]. The permeability characteristics of coal and gas outbursts are closely linked to varying gas pressures and outburst diameters [14,15]. Factors such as gas release, coal fragmentation, and changes in effective stress inevitably result in changes to the pore structure of coal [16]. Therefore, alongside stress, the microscopic pore structure of coal also directly affects the coal and gas outburst.

Coal is a kind of porous heterogeneous medium containing a large number of pore-fracture network structures [17, 18]. This intricate pore structure not only offers ample space for methane adsorption and flow channels but also plays a decisive role in the coal body itself [19,20]. The intricate nature of coal's internal pore structure plays a crucial role in comprehending how methane exists and migrates within coal formations. Numerous researchers have explored this complexity, concentrating on factors including specific surface area, pore-throat ratio, pore volume, connectivity of pores, pore ratio, and the principles of fractal geometry [21,22]. Furthermore, researchers have explored the pore structure characteristics of coal and their impact on methane adsorption capacity. Variations in pore structures among coals of different ranks have been found to influence methane adsorption properties [23–25]. Experiments using methods like LPGA-$N_2$ and LPGA-$CO_2$ have shown that micropore contribute significantly to the total pore surface area, with methane adsorption capacity increasing as micropore volume rises [26,27]. Variations in pore size distribution within coal bodies affect permeability, adsorption, desorption capabilities, and ultimately, methane extraction rates [28–30]. Although researchers have studied the

characteristics of pore structure and the methane adsorption capacity of coal under various conditions, including metamorphic degree, physical properties, moisture levels, and stress conditions, but the geological conditions of coal reservoirs, as an important factor restricting the development of coalbed methane, compared with other karst landforms, the peak cluster landforms between the peaks of the peak district is highly undulating, with a vertical height difference of hundreds of meters and steep slopes, resulting in high variability of the rock stresses overlying the coal reservoirs, which not only influences the complexity of the coal's pore structure, but also controls the gas transport channels of the coal reservoirs. Therefore, further research is needed on the complexity of pore structure in coal influenced by vertical principal stress under peak cluster landform, and understanding the coal pore structure is crucial for effectively preventing and controlling coal and gas outburst accidents.

To explore the variability of coal pore structure characteristics under different vertical principal stresses in peak-clustered topography, a combined method using MIP and LPGA-N$_2$ was employed for multi-scale quantitative characterization of coal pore structure. Fractal theory was subsequently utilized to examine the complexity and diversity of the pore structure. The study focused on the correlation between the fractal dimension and vertical principal stress, aiming to understand how the pore structure of coal behaves under varying vertical principal stresses. This study can serve as a reference for investigating the microscopic pore structure characteristics of coal reservoirs under equivalent geological conditions. Additionally, it enables mine operators to assess the gas accumulation potential in coal at different elevations, evaluate their risk of outbursts, and thereby proactively formulate effective gas control measures.

## 2. Experiment and method

### 2.1. Sample preparation

Samples of coal were gathered from the No. 9 coal layer at Longfeng Coal Mine, situated in the central part of the Upper Series of the Permian Longtan Formation, at the base of the upper part where the strata are stable and structurally simple. The No. 9 coal seam is the main economically viable seam within the mining area and is also the thickest seam, characterized by low sulfur content, making it high-quality anthracite coal. In this study, based on the characteristics of vertical variability of peak cluster landform, which is specifically manifested in the different heights between mountains, the main basis for sample selection was the height of the overlying rock layer of the coal samples, and combined with the mountain contour map of Longfeng Coal Mine and the upper and lower control maps of the mine, nine groups of coal samples under different mountain heights were selected, and the height of mountains between each group of samples ranged from 10−25 m. The coal samples were collected at the same time. These points were located at various elevations within the mine: Longfeng Coal Mine's 120910 return airflow roadway at 574m, 606m, 635m, 650m, 676m, 706m, and 120910 transport roadway at 452m, 511m, and 569m, the specific mountain elevation is shown in Table 1. The collected coal samples were screened and prepared into 80−100 mesh (0.15 mm - 0.18 mm) particles, as shown in Fig 1. For increased precision in the experiment, the coal samples were dehydrated at 105°C for a duration of 6 hours prior to commencement.

### 2.2. Experiment study

Coal pore structure encompasses pore morphology, specific surface area (SSA), connectivity, distribution of pore sizes, and other related factors. Methods for assessing pore structure include fluid-based and radiation-based techniques [31].

**Table 1. Sampling points and mountain elevation.**

| Samples | LF1 | LF2 | LF3 | LF4 | LF5 | LF6 | LF7 | LF8 | LF9 |
|---|---|---|---|---|---|---|---|---|---|
| Roadway | Transport | Transport | Airflow | Airflow | Airflow | Airflow | Airflow | Airflow | Airflow |
| | 452m | 511m | 706m | 569m | 676m | 574m | 650m | 606m | 635m |
| Mountain elevation | 1205m | 1240m | 1250m | 1270m | 1280m | 1295m | 1315m | 1335m | 1350m |

Because the measurement range of coal pore diameter is different between various methods, their applicability is also different. This research anticipates employing the pore structure segmentation technique suggested by the Soviet academic Hodort [32], that is, Micropore < 10 nm, Transition pore 10~100 nm, Mesopore 100~1000 nm, Macropore 1000~10000 nm. To determine the distribution characteristics of pores across various scales, an analysis and testing of coal pores are conducted using MIP and LPGA-N$_2$. Combined with the characteristics of the two experimental methods, macropore and mesopore are analyzed by MIP, and micropore and transition pore are analyzed by LPGA-N$_2$. In Fig 2, the classification and measurement aperture of each method for testing coal pore can be distinguished.

The MIP uses the AutoPore IV 9500 mercury intrusion instrument produced by Micromentics in the United States. The mercury injection experiment places the sample in a pressurized mercury liquid. When the pressure is low, mercury enters the cracks first; as the mercury liquid pressure increases until the pressure is greater than the capillary force of the pore throat, mercury begins to enter the pores. According to Eq. (1) Washburn's formula, by assessing the amount of mercury that is introduced into the pores of coal at varying pressures, one can derive the relationship between mercury pressure and the volume of injected mercury. This curve facilitates the examination of the related pore data according to its properties.

$$P = -\frac{2\sigma \cos \theta}{r}$$

(1)

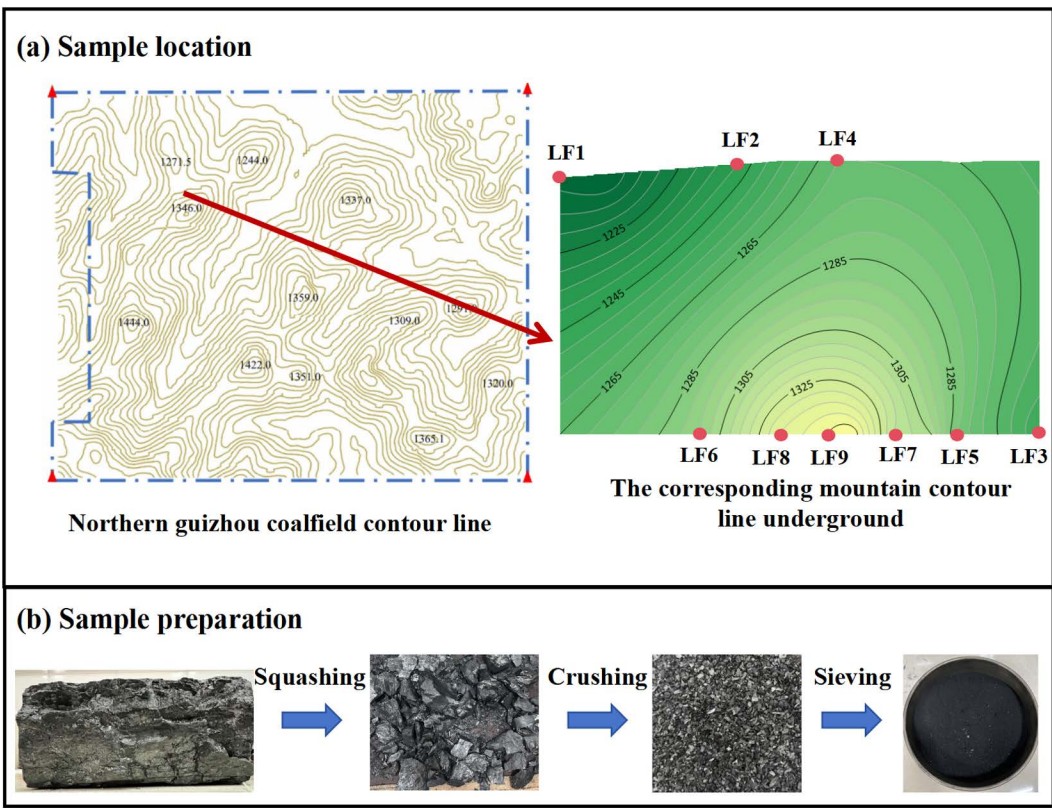

**Fig 1. The mountain contour line of the sampling point and its collection and preparation.**

In the formula: $r$ is the pore radius, m; $\sigma$ is the surface tension of mercury, N/m; $\theta$ is the contact angle between coal sample and mercury; $p$ is mercury injection pressure.

The LPGA-$N_2$ utilized the BSD-660SA3 automatic specific surface and porosity analyzer manufactured by China Best Instrument Co., Ltd. This method explores the physical adsorption characteristics of gas by coal, focusing on parameters such as pore distribution, SSA, and pore volume. Based on the Brunauer-Emmett-Teller (BET) theory of adsorption, nitrogen molecules adhere to the coal surface under the influence of the free field, but this energy alone cannot overcome molecular thermal motion. As a result, gas molecules separate from the solid surface until equilibrium is achieved where these effects balance.The adsorption and desorption processes are jointly determined by the energy of the free field and molecular thermal motion. As a result, under consistent external temperatures, the quantity of gas adsorbed changes based on the relative pressure. Analyzing the resultant curve provides insights into the pore parameters of the material. It's worth noting that due to the constraints of nitrogen adsorption experiments, this study specifically examines micropore ranging from 1–10 nm and mesopore in the range of 100–200 nm.

## 3. Expreiment and calculation results

### 3.1. Calculation of vertical principal stress

The characteristics of the peak cluster landform result in significant vertical variability in the study area, subjecting the coal bodies to different pressures from the overlying rocks. Furthermore, the principal stress in the vertical direction, which represents the pressure acting along this axis of ground stress, primarily results from the gravitational force exerted by the rocks above. Therefore, in order to quantitatively obtain the different overlying rock pressures borne by the coal body, this study examines the alteration patterns of the pore structure within the coal body influenced by the peak cluster landform, using the stress derived from the vertical principal stress equation.

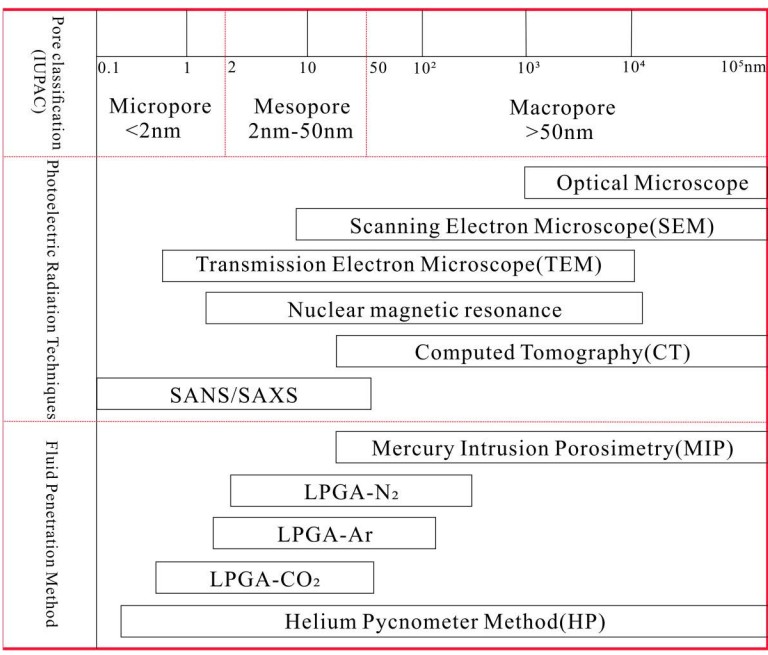

**Fig 2. Pore size classification and comparisons of the measurable ranges of testing methods.**

The formula for calculating the vertical principal stress is [33]:

$$\sigma v = \int_0^h g\rho(h)dh \tag{2}$$

In the formula: $\rho(h)$ represents the formation density in kg/m³; $g$ denotes the acceleration due to gravity, assumed as 9.8 m/s²; $h$ stands for the burial depth of the coal seam in meters.

Based on geological drilling data from Longfeng Coal Mine, the coal-rock layers above coal seam No. 9 are categorized into eight distinct layers, predominantly composed of sandstone, limestone, and mudstone. Considering the varying densities of each rock type, the calculation formula can be adjusted as follows:

$$\sigma v = \int_0^{h_1} g\rho(h_1)dh + \int_0^{h_8} g\rho(h_2)dh + \ldots + \int_0^{h_8} g\rho(h_8)dh \tag{3}$$

In the formula: $h$ represents the thickness of each coal-rock layer. Based on the geological borehole data and field measurements collected in the study area, the rock density for each layer of coal rock has been recorded, in which the average density of sandstone is 2500 kg/m³, the average density of limestone is 2650 kg/m³, the rock density of mudstone is 2300 kg/m³, and the average density of coal is 1300 kg/m³.

Utilizing Eq. (2) and (3), compute the vertical principal stresses at various elevations within the study area, and present the findings in Table 2.

### 3.2. Multi-scale pore structure characterization of coal

**3.2.1. Mercury injection test results.** The distribution of pore sizes illustrated in Figs 3 and 4 is shown through the processes of mercury injection and withdrawal. The pore structure features of coal samples vary due to differing vertical principal stresses. During mercury intrusion, the mercury injection rate for coal at different elevations generally shows a trend of rapid increase, followed by a deceleration, and then another rapid increase. When the pressure is below 0.2 MPa, the amount of mercury injected increases rapidly, indicating well-developed pores with diameters larger than 1000 nm. At pressures between 0.2 and 20 MPa, the mercury injection rate increases slowly, suggesting poor development of pores in the range of 100–1000 nm. However, when the pressure exceeds 20 MPa, the mercury injection rate increases rapidly again, indicating good development of micropore and mesopore.

Based on the "contact angle" and "ink bottle" theories, mercury can exhibit a "bottleneck" effect when it infiltrates smaller pore throats during the mercury injection procedure. Subsequently, during the mercury removal process, some mercury vapor may remain trapped in regions with poor connectivity within the pores [34,35]. The hysteresis loop identified in the mercury injection curve offers information regarding the overall pore architecture and interconnectivity of the coal [36,37]. Examination of mercury injection curves of coal from various elevations shows slight hysteresis loops at all elevations. This indicates that most pores are generally semi-closed. It suggests mesopore to macropore display some openness, whereas micropore and transition pore remain semi-open and exhibit poor connectivity.

**Table 2. Vertical geostress calculation data.**

|  | LF1 | LF2 | LF3 | LF4 | LF5 | LF6 | LF7 | LF8 | LF9 |
|---|---|---|---|---|---|---|---|---|---|
| Mountain elevation(m) | 1205 | 1240 | 1250 | 1270 | 1280 | 1295 | 1315 | 1335 | 1350 |
| Buried depth(m) | 195 | 230 | 240 | 260 | 270 | 285 | 305 | 325 | 340 |
| Stress(MPa) | 4.89 | 5.80 | 6.06 | 6.58 | 6.84 | 7.22 | 7.74 | 8.26 | 8.65 |

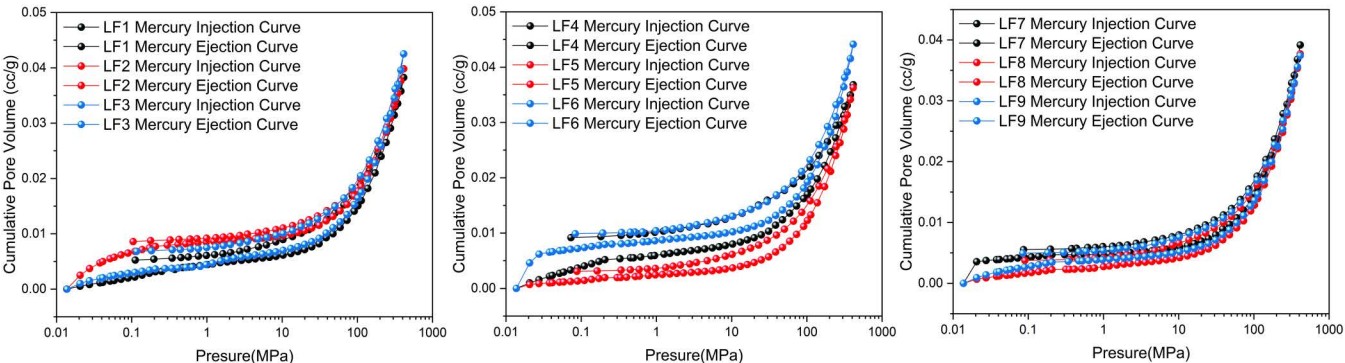

**Fig 3. Pressurized mercury curve.**

**Fig 4. Pore size distribution by MIP.**

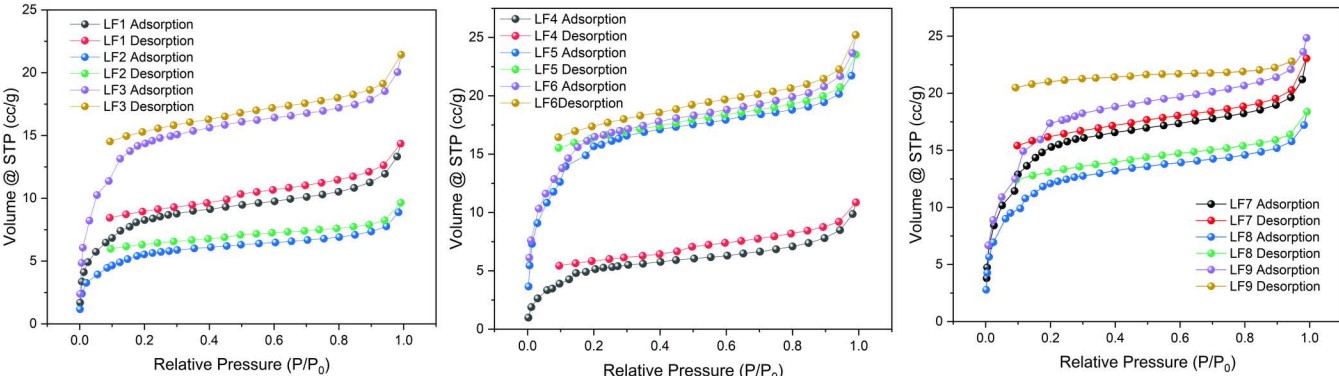

**Fig 5. LPGA-N$_2$ isotherms of coal samples.**

From Fig 3, from the left to the right, the mercury intake of different coal samples can be divided into three stages. The total mercury intake of LF1-LF3 is the largest, while the total mercury intake of LF7-LF9 is the smallest, indicating that the mercury intake of coal samples decreases with the increase of vertical principal stress. Nevertheless, at every stage, the mercury absorption in coal samples exhibits a multi-peak pattern, suggesting that the effect of vertical principal stress on mercury absorption in coal samples is incremental to some degree. Furthermore, to gain a more intuitive understanding of the pore size distribution across nine coal samples, a pore size distribution chart has been created utilizing the data from mercury injection experiments, as illustrated in Fig 4. Fig 4 clearly shows that micropore are the most abundant type of pore in the coal samples, with transition pores and macropore following in prevalence. The mesopore volume constitutes the smallest proportion, whereas the volume of micropore is roughly equivalent to the total volumes of transition and macropore combined.

### 3.2.2. Low temperature nitrogen adsorption test results.

Fig 5 displays the nitrogen adsorption isotherms, illustrating a rise in nitrogen adsorption for every coal sample as the relative pressure ($P/P_0$) increases.The nine coal samples exhibit an average nitrogen adsorption capacity of 14.93 cc/g, with the LF8 coal sample demonstrating a maximum adsorption that is 1.69 times greater. The data from the graph indicate that as the vertical principal stress increases, the effect of stress on nitrogen adsorption capacity exhibits a multi-peak variation. This suggests that under different vertical principal stresses, stress changes the degree of pore closure in the coal samples, thereby influencing their adsorption capacity.

The shape of nitrogen adsorption-desorption curve is generally used to qualitatively identify or predict the shape of coal pore structure [38–40]. When the pressure relative to the base pressure ($P/P_0$) is less than 0.2, coal samples typically exhibit a significant rise in nitrogen adsorption as $P/P_0$ increases. In the range of relative pressures from 0.2 to 0.85, the rate of nitrogen adsorption in coal samples shows a gradual increase.

According to the definition provided by the International Union of Pure and Applied Chemistry (IUPAC), the nitrogen adsorption-desorption curve illustrated in Fig 5 is recognized as a combination of the high relative pressure section of type II and the low to medium relative pressure section of type IV (a) adsorption isotherms [41]. In Fig 5, the curves representing adsorption and desorption for nine different groups of coal samples do not coincide. At relative pressures $P/P_0 < 0.45$, a noticeable hysteresis loop is still present in the coal samples, indicating a high content of micropore inside the coal where micropore adsorption occurs at lower relative pressures. The adsorption and desorption curves presented in Fig 5 indicate that all coal samples display relatively comparable hysteresis loops throughout the full spectrum of pore sizes, although certain samples demonstrate more extensive hysteresis loops compared to others which show smaller ones. The coal sample's significant hysteresis loop suggests a higher presence of slit pores or parallel plate pores, indicating

a well-developed pore structure in the coal. The coal sample exhibits a minor hysteresis loop, suggesting that a majority of the coal consists of poorly linked pores (including conical cavities, circular voids, and others). In contrast, a reduced hysteresis loop implies that the coal has a lower quantity of narrow slit-shaped pores or plate-like pores arranged parallel to poorly interconnected structures like conical or spherical pores. The results indicate that as the vertical principal stress continues to increase, the development of internal pores in the coal samples also exhibits multi-modal variations corresponding to changes in vertical principal stress.

In addition, according to the IUPAC standard, the hysteresis loops formed by the adsorption and desorption isotherms of the three samples can be classified as H2 and H5 types. It is generally believed that the inflection point of desorption branch formation is the influence of ink bottle pores. However, since the majority of coal samples did not exhibit a turning point in the desorption branch, it is considered that the ink bottle-shaped pores are less abundant in these coal samples. Generally, the nitrogen adsorption-desorption curve does not exhibit a closed hysteresis loop due to the expansion and inflation of the coal body's pores throughout the adsorption process [42], which is common in relatively developed pores [43].

The analysis of cumulative and differential pore volumes shown in Fig 6 depicts the distribution of pore sizes. It is significant to note that most coal samples demonstrate a multi-peak characteristic in their pore size distribution. The peaks of the other samples are predominantly between 1.5–4 nm, and the main peak for most coal samples is around 2 nm, The difference of pore volume in coal samples is mainly reflected in the pores of 1.5–20 nm.

### 3.2.3. Combined characterization of coal pore structure

The mercury intrusion experiment measures pore sizes by applying mercury pressure to the coal sample. Initially, mercury infiltrates cracks in the sample and gradually penetrates into the pores as pressure increases. As pressure keeps increasing, certain pores fail to endure the mercury pressure, which causes multiple cracks to form and leads to variations in the experimental results. Conversely, the sizes of pores are ascertained through low-temperature liquid nitrogen adsorption, where condensed liquid nitrogen fills the pores. Liquid nitrogen first enters small pores and subsequently fills larger ones.

Since the mercury pressure experiment is based on the pressure of mercury liquid to measure the pore size, at the beginning of the experiment, the mercury liquid was first pressed into the cracks of the coal samples, and then gradually pressed into the pores with the increase of pressure. However, as the pressure increases, some pores cannot resist the pressure of mercury liquid to form a large number of fissures, resulting in experimental deviation, and limited by the surface tension and contact angle of mercury, resulting in extremely low efficiency of mercury intrusion into micro and small pores; low-temperature liquid nitrogen adsorption is based on the cohesion of liquid nitrogen to fill the pores to detect the pore size, and is able to accurately analyze the micro and small pores, but has low sensitivity to the characterization of the medium and large pores, mainly because when measuring large pore sizes, the liquid nitrogen cannot coalesce.

The main reason is that when measuring the large pore size, the liquid nitrogen can not be coalesced with the increasing pressure, which makes the large pore size data of the measured coal samples have a large deviation. Therefore, characterizing the pore structure of coal by combining the two experiments not only realizes the full-scale coverage of pores and makes up for the blind spot of pore diameter detection of a single method, but also provides a more comprehensive and reliable data basis for the characterization of the pore structure of coal through the synergistic analysis of the morphology, specific surface area, and fractal features obtained from the two experiments. In addition, because the pore structure of coal can be partially overlapped when analyzing the pore structure of mercury pressure and low-temperature nitrogen adsorption experiments, resulting in a large error in the experimental data, based on the different sensitivities of the results of the two experiments for the characterization of multi-scale pores, a critical pore diameter was established as a pore diameter indicator to distinguish between the results of mercury pressure experiments and low-temperature nitrogen adsorption experiments, and the critical pore diameter was determined as the intersection point $r_c$ of the change curves of the measured pore diameters and the pore volumes of the two experiments.

**Fig 6. Pore size distribution characteristics obtained by LPGA-N$_2$.**

The critical pore size was determined mainly as the intersection of two experimentally determined pore size versus pore volume curves $r_c$. If the diameter of the pore is less than $r_c$, data from the adsorption experiment with liquid nitrogen at low temperatures is utilized. Conversely, when the diameter of the pore is larger than $r_c$, the data from the mercury intrusion experiment is utilized. Since some coal samples exhibit multiple intersections of the curves derived from both experiments, the last intersection point is selected as the critical pore diameter $r_c$ for that particular coal sample. The determination of the critical pore size guarantees the accuracy of the pore structure analysis in the combination of mercury pressure experiments and low-temperature nitrogen adsorption experiments. Presented in Fig 7 and Table 3 are the critical pore diameters determined via graphical analysis for the coal samples.

Fig 8 demonstrates that, by employing the critical pore sizes acquired, one can generate the pore size distribution graph derived from experiments that include mercury intrusion porosimetry and nitrogen adsorption techniques. The volumetric ratios for each pore size are presented in Table 4. Compared to the pore size distribution from the single mercury intrusion experiment, there is a significant change in the proportion of mesopore and macropore in the combined characterization with mercury intrusion and nitrogen adsorption, indicating that mercury intrusion analysis inaccurately assesses micro and mesopore. Furthermore, for coal samples with significant nitrogen adsorption, the proportion of micropore

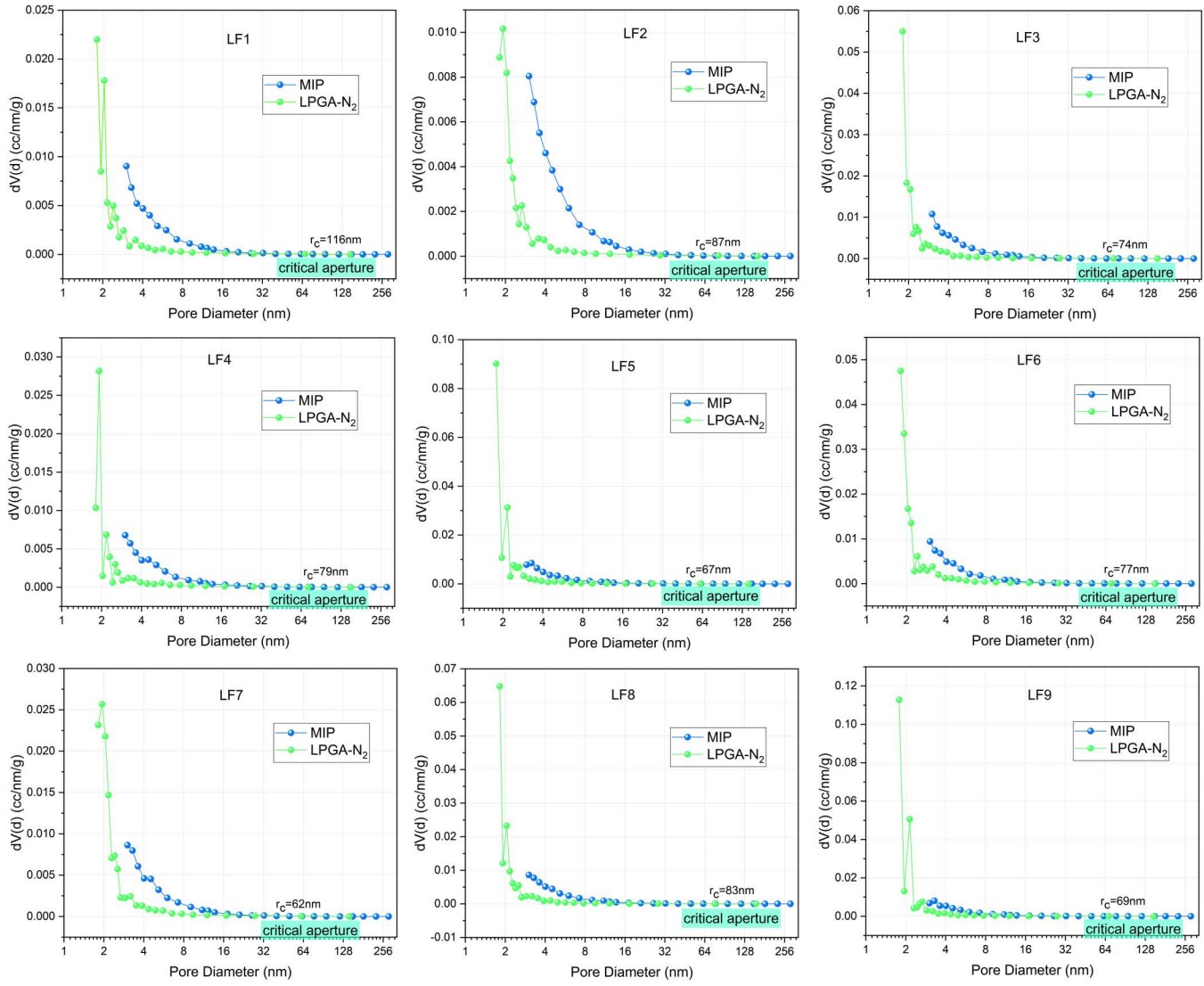

**Fig 7. Determination of critical aperture.**

**Table 3. Critical pore size values of each coal sample.**

| Samples | LF1 | LF2 | LF3 | LF4 | LF5 | LF6 | LF7 | LF8 | LF9 |
|---|---|---|---|---|---|---|---|---|---|
| Critical pore diameters (nm) | 116 | 87 | 74 | 79 | 67 | 77 | 62 | 83 | 69 |

exceeds 60% of the total pore volume. Similar to the previous analyses, as the vertical principal stress increases, the proportion of micropore volume in the total pore volume also shows multi-peak variations. In summary, compared to the single use of mercury intrusion for pore size distribution characterization, which is more suitable for testing mesopore to macropore but overlooks the accuracy of micropore and transition pore characterization. As a result, merging the two experiments allows for a more precise description of the coal's multi-scale pore structure.

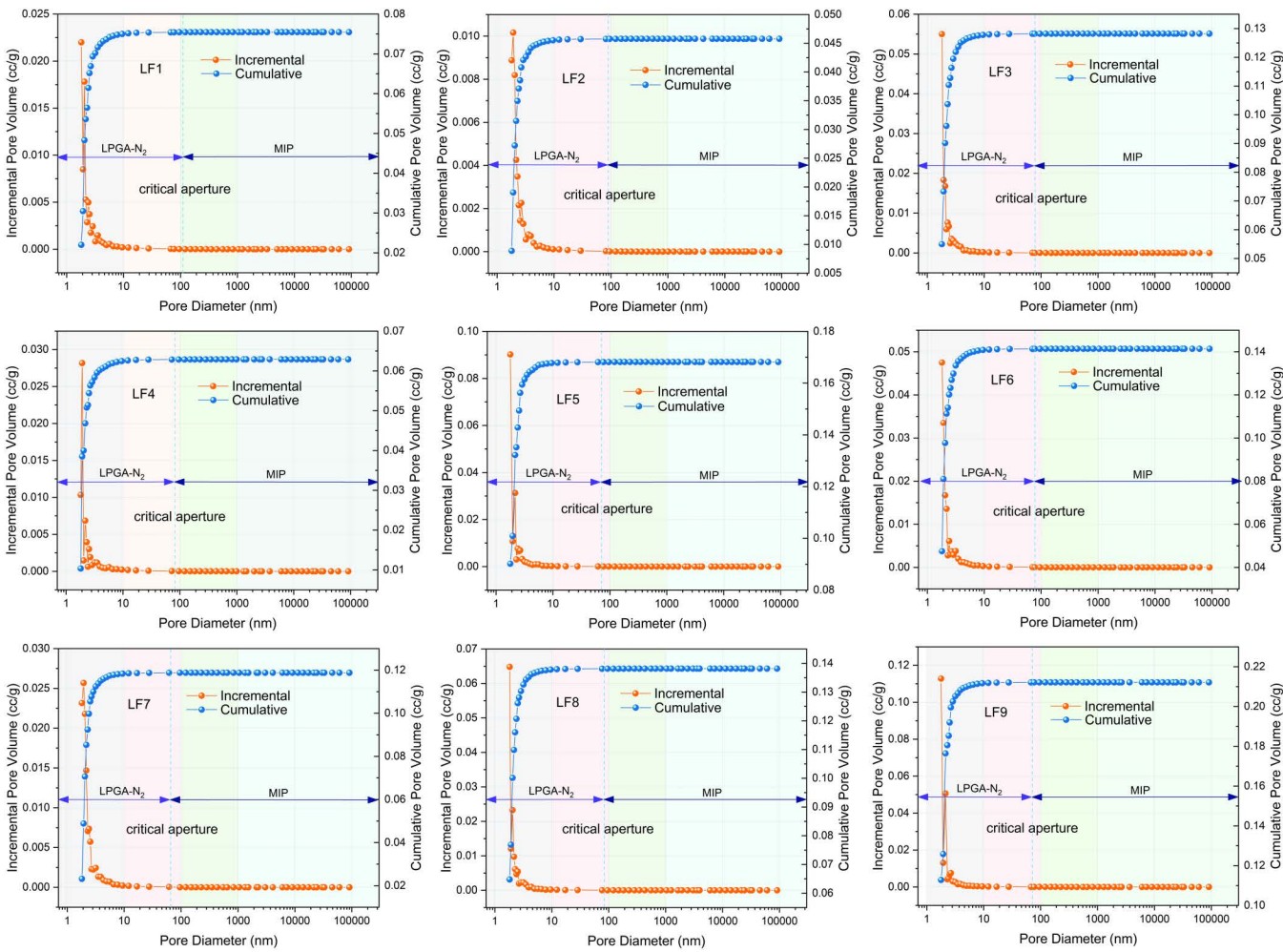

**Fig 8. Joint characterization of mercury intrusion and nitrogen adsorption.**

**Table 4. Pore size distribution of coal samples.**

| Coal samples | Pore volume/(cc/g) | | | | | Pore volume ratio/% | | | |
|---|---|---|---|---|---|---|---|---|---|
| | Micro pore | Transition pore | Meso pore | Macro pore | total | Micro pore | Transition pore | Meso pore | Macro pore |
| LF1 | 0.00456 | 0.00157 | 0.00476 | 0.01186 | 0.02276 | 52.12 | 20.92 | 6.91 | 20.05 |
| LF2 | 0.00861 | 0.00107 | 0.00354 | 0.00727 | 0.02049 | 35.47 | 17.28 | 5.23 | 42.02 |
| LF3 | 0.00470 | 0.00229 | 0.00543 | 0.01947 | 0.03189 | 61.05 | 17.03 | 7.18 | 14.73 |
| LF4 | 0.00594 | 0.00207 | 0.00504 | 0.00977 | 0.02282 | 42.81 | 22.07 | 9.08 | 26.04 |
| LF5 | 0.00251 | 0.00118 | 0.00548 | 0.02188 | 0.03105 | 70.47 | 17.64 | 3.81 | 8.09 |
| LF6 | 0.00876 | 0.00146 | 0.00691 | 0.02105 | 0.03818 | 55.14 | 18.11 | 3.82 | 22.94 |
| LF7 | 0.00480 | 0.00114 | 0.00596 | 0.01822 | 0.03012 | 60.49 | 19.78 | 3.79 | 15.94 |
| LF8 | 0.00285 | 0.00137 | 0.00463 | 0.01716 | 0.02601 | 65.98 | 17.82 | 5.27 | 10.94 |
| LF9 | 0.00395 | 0.00122 | 0.00556 | 0.02783 | 0.03856 | 72.17 | 14.41 | 3.17 | 10.26 |

## 3.3. Fractal feature analysis

The irregularity and complexity of coal's pore structure pose challenges for conventional quantification methods [44]. Fractal theory, pioneered by B.B. Mandelbrot, offers an effective approach to studying and describing coal pore morphology [45,46]. This theory, extensively used in analyzing the surface properties of self-similar materials [47], employs the fractal dimension ($D$) to elucidate the complexity and heterogeneity of coal pores. A higher fractal dimension indicates greater irregularity in pore shape and rougher surface texture [48]. To ensure accuracy in characterizing the fractal properties of coal samples, the fractal characteristics of mesopore to macropore were determined using mercury intrusion data, while micropore and transition pore were analyzed using low-temperature nitrogen adsorption experiments.

The FHH (Frenkel-Halsey-Hill) model was used to calculate the fractal dimension of micropore and transition pore in coal [49,50]. The calculation equation of the model is as follows:

$$\ln\left(\frac{V}{Vm}\right) = (D-3)\ln\left[\ln\left(\frac{P0}{P}\right)\right] + C$$

(4)

In the formula: $V$ is the amount of nitrogen adsorption, mL/g; $V_m$ is the volume of single molecule adsorbed gas, mL/g; $P_0$ is the saturated vapor pressure of gas adsorption, MPa; $P$ is nitrogen adsorption pressure, MPa; $C$ is a constant; $D$ is the fractal dimension.

Based on Eq. (4) the data related to $\ln(V/V_m)$ and $\ln[\ln(P_0/P)]$ are fitted, as shown in Fig 9, the slope of the fitted line segment in the graph is the fractal dimension value. The data is separated from the point $P/P_0 = 0.5$, and the fractal dimension of the section $P/P_0 < 0.5$ is set as $D_1$, which represents the surface roughness of the coal pores. At this point, the capacity for coal gas adsorption is significantly influenced by van der Waals forces [51]. Certainly, here is a revised version of your text with the necessary modifications to avoid duplication:The value of $D_2$ is assigned to the fractal dimension of the section where $P/P_0 > 0.5$. This aspect reflects the complexity of the pore structure found in coal. A more complex pore arrangement is associated with improved capacity for gas adsorption [52].

The Menger model is applied to compute the fractal dimension for mercury injection experiments. The Menger model, crafted specifically for mesopore and macropore, establishes the fractal dimension by examining the correlation between the pressure of inlet mercury and the volume of the pores [53]. Menger model for mesopore and macropore fractal dimension $D$ is calculated as shown in Eq.(5).

$$D = 4 + lg\left(\frac{dV}{dp}\right)/lg\,p$$

(5)

In the formula: $p$ is the applied pressure in the process of mercury injection, MPa; $V$ is the pore volume at mercury pressure p, mL/g; $D$ is the fractal dimension of mesopore and macropore, dimensionless.

In Fig 9, the relationship between $\ln(V/V_m)$ and $\ln[\ln(P_0/P)]$ is depicted for three coal samples. A scatter diagram is utilized for fitting calculations, revealing an average fitting coefficient of 0.94 in the low-pressure range ($P/P_0 < 0.5$). The impressive accuracy suggests that the FHH model effectively examines the fractal characteristics of micropores and transitional pores in the chosen coal samples.

Fig 10 presents a scatter diagram following logarithmic transformation and fitting analysis of pressure versus the ratio of pore volume to pressure increment. The Menger model effectively calculates the fractal properties of mesopore and macropore within the chosen coal samples, as evidenced by an average fitting coefficient of 0.903.

The fractal dimensions of micropore, transition pore, mesopore, and macropore for nine coal samples (LF1... LF9), affected by varying vertical principal stresses, were calculated. The fractal dimension serves to characterize the complexity of coal pore structures present in porous substances. A higher fractal dimension indicates an elevated degree of

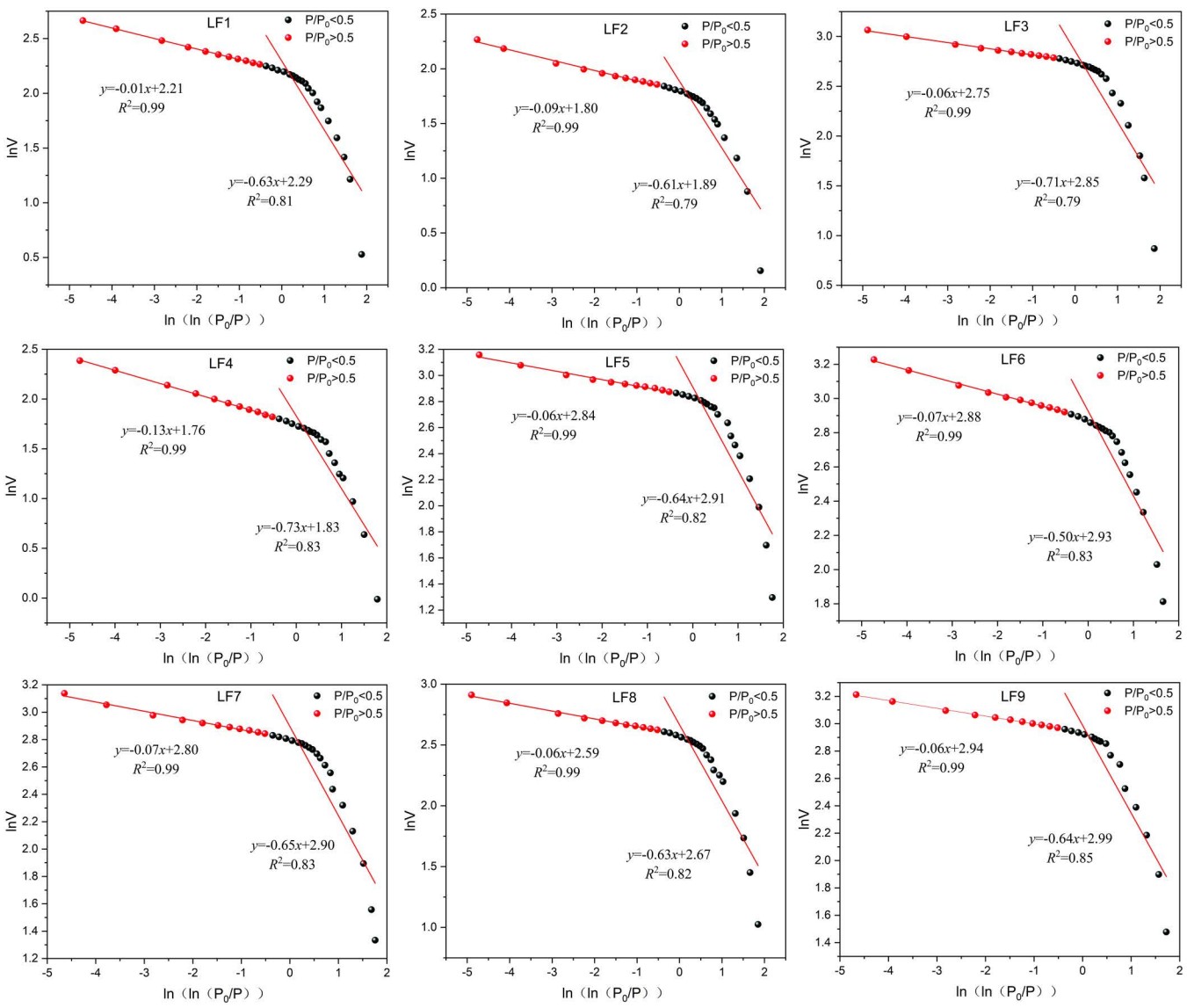

**Fig 9. The relationship between micropore and transition pore ln (ln ($P_0/P$)) and ln$V$.**

complexity and irregularity within the pore architecture. Generally, fractal dimensions lie between 2 and 3. A dimension nearing 2 implies reduced surface roughness and more straightforward pore volume configurations. In contrast, a dimension approaching 3 signifies increased surface roughness and more intricate pore volume arrangements. Fractal dimensions exceeding 3 often result from inherent macropore or extensive internal cracks within the coal body, beyond the scope of fractal theory analysis [54,55].

The fractal dimension value of each coal sample is shown in Table 5. In the nine types of coal samples studied, the fractal dimension of micropore and transition pore generally ranges between 2 and 3. However, in the low-pressure regime ($P/P_0$<0.5), this dimension tends towards 3, suggesting a smoother surface for micropore and transition pore across various coal samples, yet with a structurally complex pore network exhibiting strong heterogeneity. Furthermore, the fitting coefficient on the left side of the fractal dimension plot for micropore and transition pore exceeds that on the

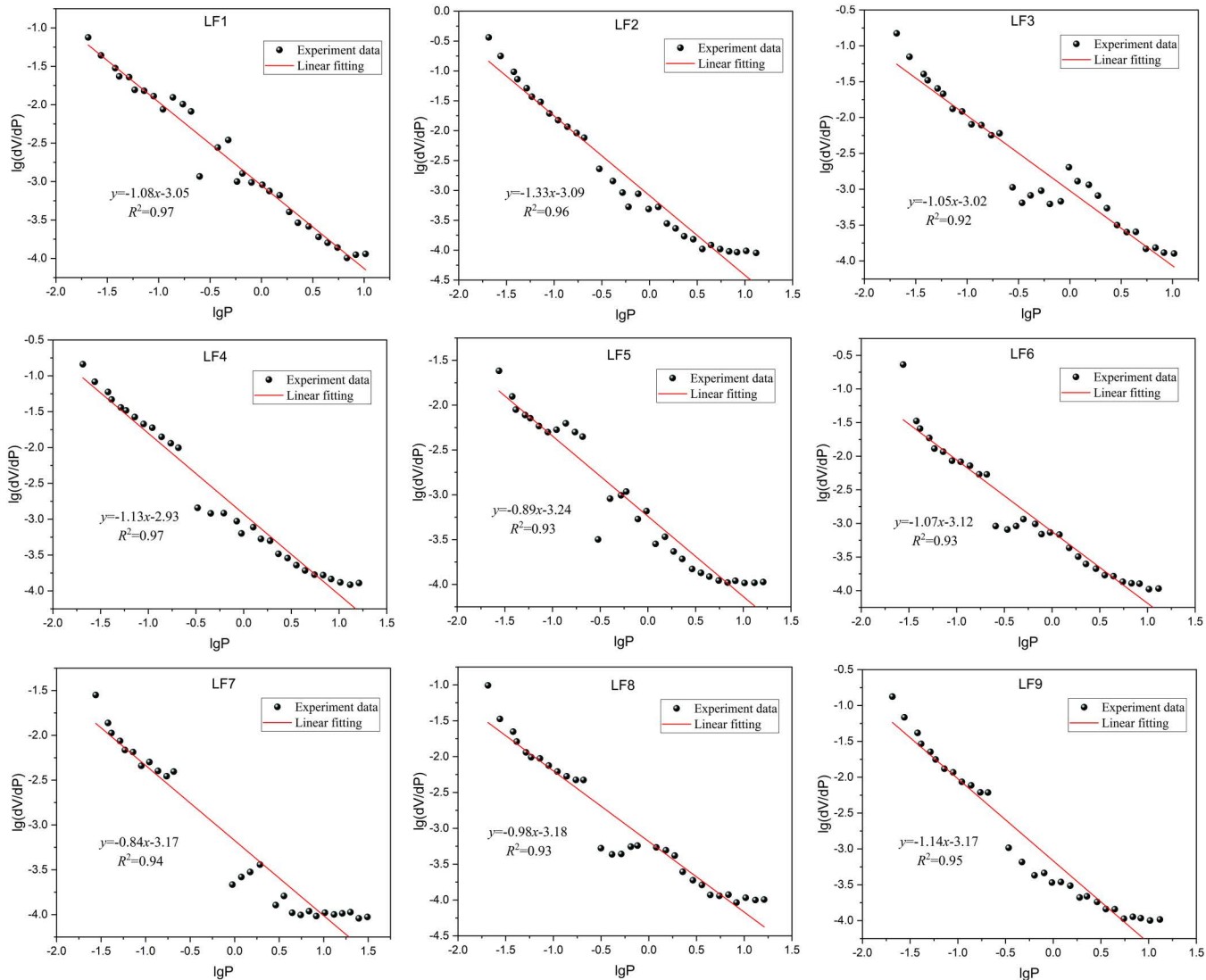

**Fig 10. The relationship between mesopore and macropore lg*P* and lg(d*V*/d*P*).**

right side, indicating that pore structure complexity significantly influences pore adsorption capacity. In contrast to micropores and transition pores, the surfaces of mesopores and macropores are more textured; however, their pore structure's complexity and heterogeneity are marginally less than those found in micropores and transition pores. Overall, the pore architecture of the coal samples reveals a variety of fractal characteristics. As the vertical principal stress rises, the coal samples exhibit a more diverse pore structure.

## 4. Analysis and discussion

### 4.1. The relationship between vertical principal stress and specific surface area, pore volume

Based on previous studies, this paper defines pore size greater than 100nm as seepage pore and pore size less than 100nm as adsorption pore [56]. Coal gas adsorption is dominated by physical adsorption, which occurs on the coal

Table 5. Calculation of fractal dimension of multi-scale pores in coal samples.

| Samples | Micropore and Transition pore | | | | Mesopore and Macropore | |
|---|---|---|---|---|---|---|
| | Goodness-of-fit ($P/P_0 < 0.5$) | Fractal dimension ($P/P_0 < 0.5$) | Goodness-of-fit ($0.5 < P/P_0$) | Fractal dimension ($0.5 < P/P_0$) | Goodness-of-fit | Fractal dimension |
| LF1 | 0.81 | 2.37 | 0.99 | 2.99 | 0.97 | 2.92 |
| LF2 | 0.79 | 2.39 | 0.99 | 2.91 | 0.96 | 2.67 |
| LF3 | 0.79 | 2.29 | 0.99 | 2.94 | 0.92 | 2.95 |
| LF4 | 0.83 | 2.27 | 0.99 | 2.87 | 0.97 | 2.87 |
| LF5 | 0.82 | 2.36 | 0.99 | 2.94 | 0.93 | 3.11 |
| LF6 | 0.83 | 2.5 | 0.99 | 2.93 | 0.93 | 2.93 |
| LF7 | 0.83 | 2.35 | 0.99 | 2.93 | 0.94 | 3.16 |
| LF8 | 0.82 | 2.37 | 0.99 | 2.94 | 0.93 | 3.02 |
| LF9 | 0.85 | 2.36 | 0.99 | 2.94 | 0.95 | 2.86 |

surface, and gas molecules are more easily adsorbed on the pore surface when the pores have a high specific surface area. In addition, the gas mainly permeates in the form of laminar or turbulent flow in the macropore and mesopore, and exists in the form of capillary condensate or physical adsorption film on the pore wall in the micropore and transition pore, which indicates that the seepage pore mainly serves as a gas transport channel and the seepage pore has a small specific surface area, which makes a limited contribution to the amount of adsorption, while the adsorption pore has a very high specific surface area in the pore, accounting for more than 80% of the total specific surface area, which provides a large number of adsorption sites for the gas molecules provide a large number of adsorption sites, which directly determines the amount of coal gas adsorption [57,58]. Therefore, in order to study the differences in the pore structure of coal under the influence of vertical principal stress, this section mainly focuses on the variation rules of specific surface area and pore volume of adsorption pores with vertical principal stress. By integrating results from MIP and LPGA-N$_2$, the trend diagram of the relationship between vertical principal stress and SSA, pore volume can be obtained.

Fig 11 presents that it is evident that the SSA and pore volume of adsorption pores exhibit multiple peaks in response to changes in vertical principal stress. The trend diagram of vertical principal stress and specific surface area reveals three distinct peak levels.The first peak reaches its maximum at a vertical principal stress of 6.06 MPa, with a SSA of 36.26 m$^2$/g. The minimum value occurs at 5.41 MPa, yielding a SSA of 4.21 m$^2$/g. This represents a 1.12 times difference in vertical principal stress and an 8.61 times difference in specific surface area.The second peak's maximum occurs at a vertical principal stress of 6.84 MPa, corresponding to a SSA of 40.51 m$^2$/g. The minimum is observed at 6.45 MPa, resulting in a SSA of 0.48 m$^2$/g. This reflects a 1.06 times difference in vertical principal stress and an 84.39 times difference in specific surface area.The third peak's maximum is recorded at a vertical principal stress of 8.65 MPa, with a SSA of 52.44 m$^2$/g. The minimum value is at 7.48 MPa, yielding a SSA of 0.11 m$^2$/g. Here, the vertical principal stress varies by 1.15 times.

The trend diagram of vertical principal stress and adsorption pore volume exhibits three distinct peak levels. The vertical principal stress exhibits a variation of 1.04 times between its maximum and minimum values during the initial peak, while the SSA of the adsorption pore shows a difference of 2.83 times. Similarly, in the second peak, the disparity in the maximum and minimum vertical principal stress values remains at 1.04 times, and the difference in the SSA of the adsorption pore is 2.38 times.The difference in the vertical principal stress between the maximum and minimum values at the third peak is 1.04 times, while the variation in the SSA of the adsorption pore is 1.67 times. Similar to the adsorption pore, the volume of this pore along with the vertical principal stress shows three separate peak levels. At each of these three peaks, the difference between the maximum and minimum values of the vertical principal stress is observed to be 1.04, 1.04, and 1.05 times, respectively. This finding aligns with the previously mentioned conclusion. Conversely, the

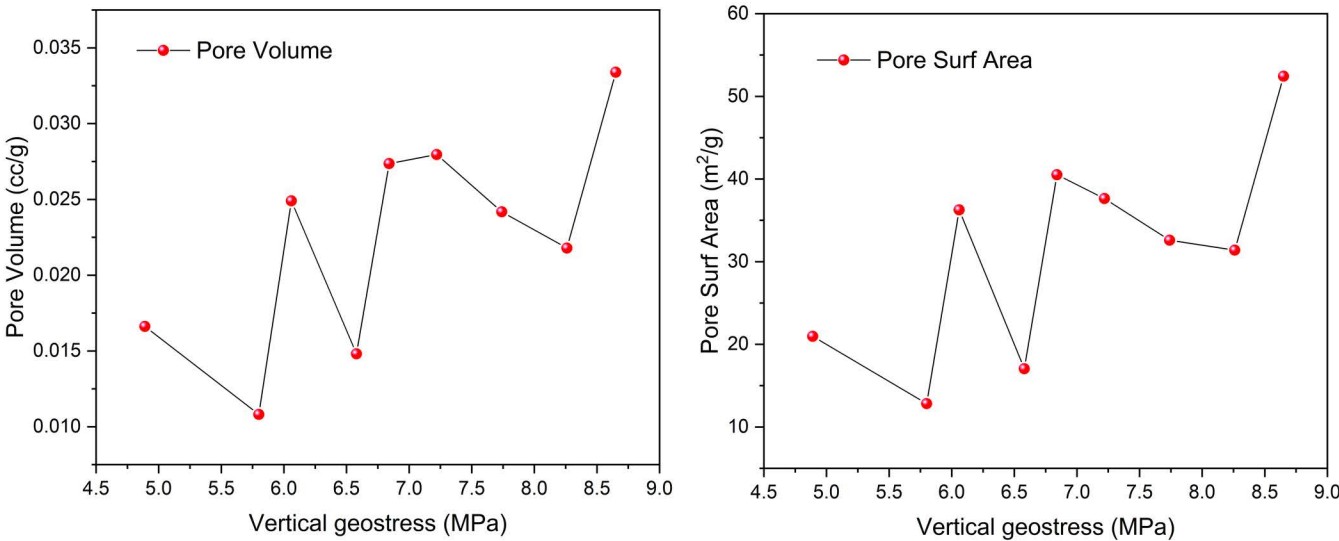

**Fig 11. Trend chart of vertical principal stress and pore structure parameters.**

variation in the maximum and minimum values of the adsorption pore volume is recorded as 2.30, 1.60, and 1.53 times, respectively.

The analysis above illustrates that variations in the vertical principal stress lead to multi-peak alterations in both the SSA and pore volume of the adsorption pores. As the peak levels progress, the extent of change among the peak values diminishes. Additionally, an increase in the vertical principal stress leads to a reduction in both the SSA and the pore volume of the adsorption pores associated with the peak levels. According to the experimental results, the vertical principal stresses show a relatively symmetric peak phenomenon with respect to the pore structure and fractal dimension, which cannot be reflected by basic functions such as polynomial and exponential functions, for example, while the Gaussian function can naturally fit this statistical property, and its bell curve can effectively describe the concentration trend and the degree of discretization of the variables around the peak value. To better elucidate the multi-peak level variations between vertical principal stress and the SSA/pore volume of the adsorption pore, Gaussian functions are employed. Specifically, the Gaussian function expression is as follows:

$$y = y_0 + A/(w\sqrt{\frac{p_i}{4\ln 2}})e^{\frac{-4\ln 2(x-x_C)^2}{w^2}}$$

(6)

Based on Eq (6), the vertical principal stress is fitted to the adsorption pore SSA and pore volume, and the fitting plot is shown in Fig 12. The coefficients that fit the vertical principal stress in relation to SSA and pore volume are 0.8529 and 0.8833, respectively, which suggests a strong correlation. While the RMSE value of pore volume was 0.0027 cc/g, the RMSE value of specific surface area was 4.5985 $m^2$/g, indicating that the average deviation of the model predicted values from the measured values were both less than 12% of the range of the measured values. In addition, the residual analysis plots for adsorption pore SSA and pore volume are shown in Fig 13. The residual analysis of pore volume showed that the residuals showed a random distribution with the variation of vertical principal stresses, with no obvious trend or regular pattern, indicating that the Gaussian function model adequately described the main variations in the data, and no obvious systematic bias was found. While the residual analysis plot of the pore specific surface area showed a rising trend during the growth of the vertical principal stress, indicating that the Gaussian model may not be able to fully capture the subtle

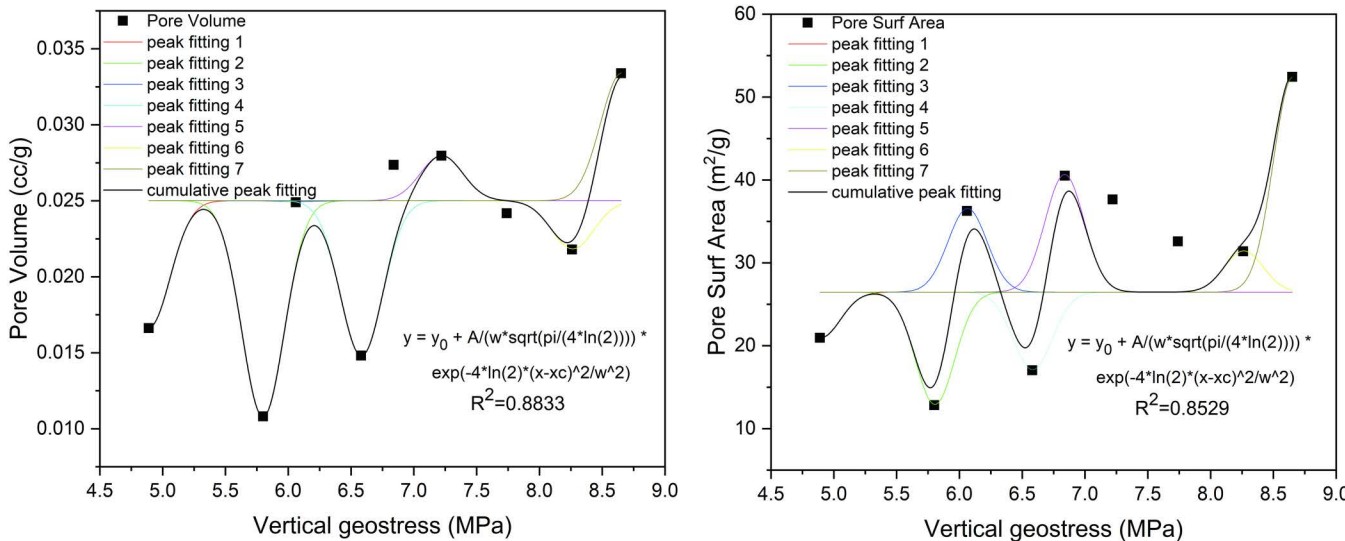

**Fig 12. Fitting diagram of vertical principal stress and pore structure parameters.**

features of the pore structure changes in this stress interval, which may be related to the special changes in the specific surface area structure of the coal samples at this stage. Nevertheless, the overall goodness-of-fit index indicates that the Gaussian model is still an effective tool to describe the overall trend of this study.

### 4.2. The relationship between vertical principal stress and pore structure complexity

To some extent, changes in vertical principal stress influence the pore structure of coal samples. As demonstrated in previous sections, the fractal dimension provides a quantitative measure of the complexity of these pore structures, suggesting a correlation between vertical principal stress and fractal dimension. In order to further clarify the relationship between

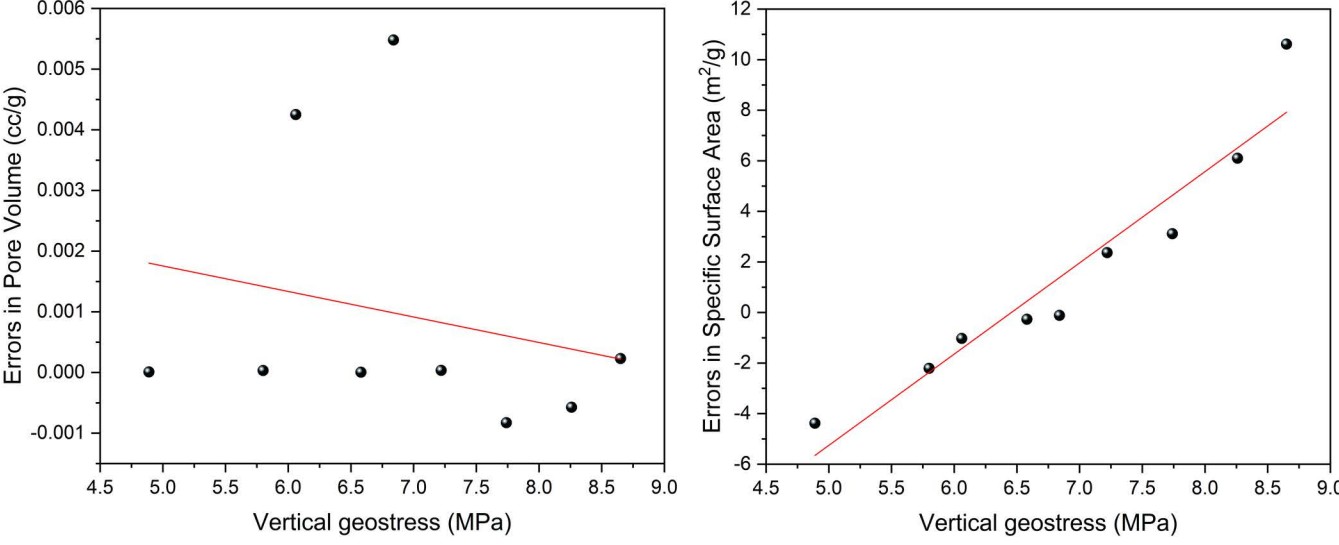

**Fig 13. Residual analysis plot of pore volume and specific surface area.**

vertical principal stress and pore structure complexity, based on the previous content, the fractal dimension $D$ is utilized to represent the pore structure complexity. The fractal dimension $D_1$ is taken into account as it reflects the roughness of the pore surface; a higher $D_1$ value suggests that the pore surface is rougher. Additionally, the fractal dimension $D_2$, which represents the inhomogeneity of the pore structure, is also examined, with a greater $D_2$ value indicating a more pronounced inhomogeneity in the pore structure. To ensure consistency in the fractal dimension of adsorption pores relative to changes in vertical principal stress, the Gaussian function previously utilized is employed to analyze their relationship, as depicted in Fig 14.

Fig 14 illustrates that the curve fitting coefficient between the vertical principal stress and the $D_1$ value is 0.7799, indicating a strong fitting relationship between these two variables. As the vertical principal stress increases, the $D_1$ value exhibits a multi-peak trend, suggesting that the surface roughness of the pore surface in the coal sample initially decreases, then increases, subsequently decreases again, and finally increases once more. In contrast, the curve fitting coefficient between the $D_2$ value and vertical principal stress is only 0.4684, reflecting a single-peak 'U'-shaped upward concave trend. This indicates that, with increasing vertical principal stress, the complexity of the pore structure in the coal samples first decreases and then increases. Moreover, the analysis of the correlation between the vertical principal stress and the $D_1$ and $D_2$ values indicates that the vertical principal stress primarily affects the roughness of the pore surfaces in the adsorption pores of the coal sample, with an increase in roughness observed as the vertical principal stress rises.

Moreover, micropore within coal represent the primary reservoir for gas storage. The pore volume and SSA of adsorption pores serve as indicators of the coal's gas adsorption capacity, while the fractal dimension provides a quantitative measure thereof. Fig 15 illustrates the connection between the characteristic parameters of adsorption pores and the fractal dimensions $D_1$ and $D_2$.

Fig 15 illustrates that the pore structure characteristics of coal samples exhibit a ternary linear relationship with the fractal dimension. The relationship between $D_1$, $D_2$, specific surface area, and pore volume resembles an 'S' type curve, albeit with opposing shapes for the two parameters. Specifically, the $D_1$ value initially decreases and then increases as pore volume and SSA rise, while the $D_2$ value demonstrates the opposite trend, increasing initially before decreasing with the same parameters. This indicates that as the pore volume and SSA of coal samples increase, the pore surface roughness and structural complexity first increase and then decrease. Furthermore, given that vertical principal stress significantly

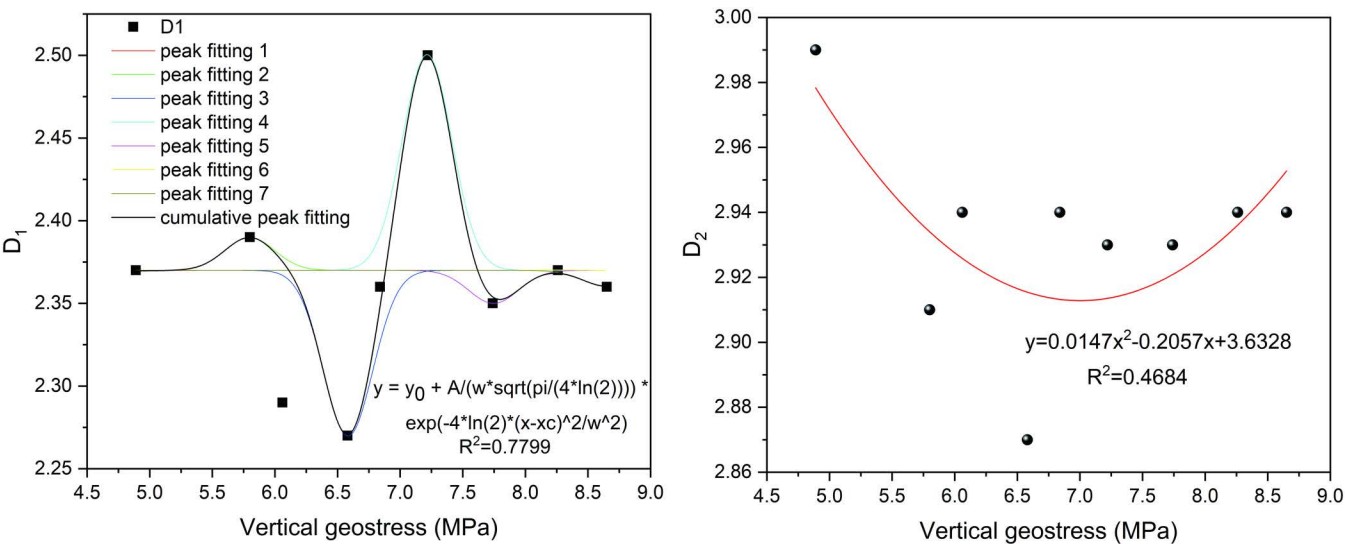

**Fig 14. The relationship between vertical geostress and fractal dimension.**

                                    

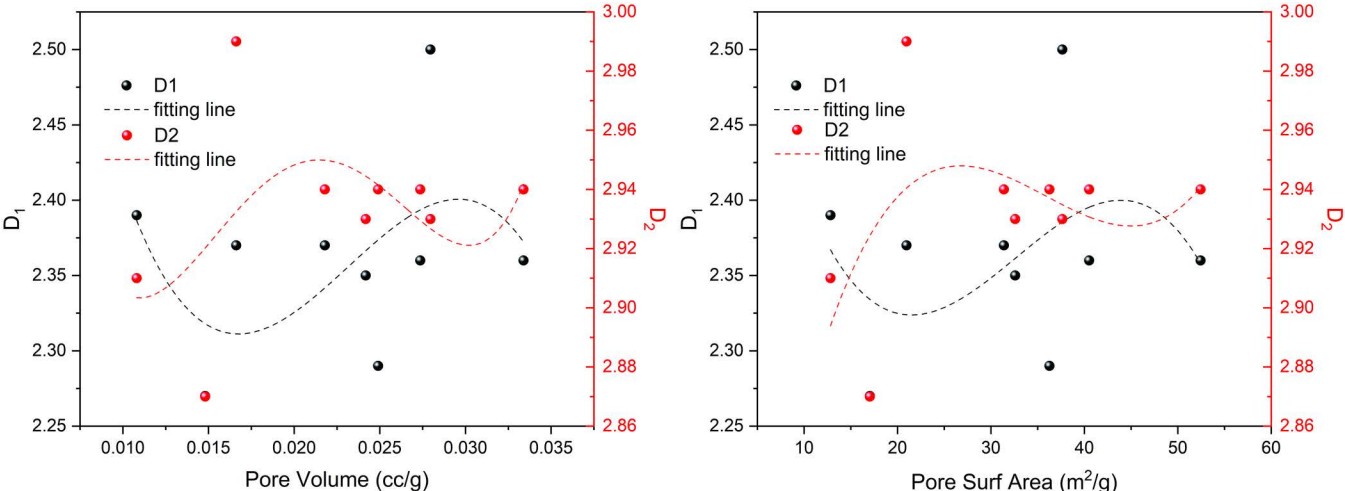

**Fig 15. The relationship between pore structure parameters and fractal dimension.**

influences pore surface roughness, highlighting the factors that influence the adsorption capacity of coal samples is crucial. The examination of the connection between pore volume, specific surface area, and the $D_1$ value depicted in Fig 15 indicates that variations in the $D_1$ value are primarily driven by pore volume. Consequently, It can be concluded that the capacity for gas adsorption in coal samples mainly relies on the roughness of the pore surface and the volume of the pores.

Based on the analysis presented, there are notable differences in the relationship between vertical principal stress and pore surface roughness. Specifically, with the progressive increase in vertical principal stress, certain adsorption pores become closed, while numerous micro-cracks are generated due to the applied stress. This phenomenon impacts the development of adsorption pores and can result in the formation of semi-closed pores influenced by the presence of micro-cracks. Consequently, the degree of pore opening varies, which in turn affects the adsorption pore content and specific surface area of the coal sample, leading to alterations in the complexity of the pore structure and the roughness of the pore surfaces. Therefore, it can conclude that vertical principal stress influences factors associated with pore structure, such as the specific surface area and pore volume of coal samples, which subsequently affects the roughness and complexity of the pore structure. These changes could result in an alteration of the coal samples' adsorption capacity.

The geological conditions of coal reservoirs are an important factor in the development of CBM, and as a key component of the karst geomorphology system, the coal under this geomorphology has obvious vertical variability, which makes the pore structure of the coal more complex, and leads to the difficulty in clarifying the law of gas enrichment under this geomorphology. Therefore, the study of the multi-scale pore structure characteristics of coal and its complexity under the peak cluster landform can further provide reference for the study of the gas storage and enrichment law, which can not only deepen the gas storage law under special geological conditions, but also lay the foundation for the key technologies of resource development, disaster prevention and control in the similar topography, so as to guarantee the safety of mine mining.

## 5. Conclusion

In this paper, the combination of high-pressure mercury pressure and low-temperature nitrogen adsorption testing method was used to quantitatively and comprehensively characterize the change rule of coal microporous structure under the influence of vertical height difference characteristics of the peak cluster landform, and based on the fractal theory, the

fractal dimensions of the seepage pores and adsorption pores of the coal were respectively calculated, and then the change of the coal pore structure characteristics and its complexity under the peak cluster landform was investigated. The following conclusions were obtained:

(1) Influenced by the vertical elevation difference characteristics of the peak cluster landform, the coal is subjected to a gradual increase in vertical principal stresses in the overlying rock strata as the elevation of the mountain increases. Unlike individual techniques, the combined use of mercury injection and low-temperature nitrogen adsorption experiments provides better accuracy and thoroughness in analyzing the multi-scale pore structure of coal. The pore content obtained from this combined approach shows clear multi-peak variations that correspond to fluctuations in vertical principal stress.

(2) The pore volume and SSA of adsorption pores in each coal sample under varying vertical principal stresses were analyzed using Gaussian function fitting. The fitting coefficients for vertical principal stress with SSA and pore volume of adsorption pores were 0.8259 and 0.8833, respectively, indicating effective Gaussian function fitting. The SSA and pore volume of the adsorption pores in each coal sample exhibit a multi-peak variation in response to changes in vertical principal stress. The degree of variation between the peak levels diminishes as the peak levels progress, and with an increase in vertical principal stress, the corresponding adsorption pores between the peak levels also change.

(3) The fractal dimensions $D_1$ and $D_2$ of adsorption pores show multi-modal variations in response to changes in vertical principal stress. The surface roughness of adsorption pores in coal samples is mainly determined by vertical principal stress, while gas adsorption capacity is primarily influenced by pore surface roughness and pore volume. The vertical principal stress significantly influences the development of micro-cracks within the pores of coal samples, thereby affecting the characteristics and complexity of their pore structure. This, in turn, leads to alterations in the gas adsorption capacity of the coal samples.

## Supporting information

**S1 File. Supporting information.**
(DOCX)

## Acknowledgments

First of all, we are grateful to the College of Mining of Guizhou University for giving us high-precision instruments and a good experimental environment. And, we also thank the other members of the team for the help and support provided to this article, so that this article can be successfully completed.

## Author contributions

**Conceptualization:** Jin Wang.

**Data curation:** Jin Wang.

**Investigation:** Hong Lan, Zhonglin Chen, Bo Li.

**Project administration:** Lulin Zheng.

**Resources:** Lulin Zheng.

**Supervision:** Yujun Zuo, Yiping Zhang.

**Validation:** Yujun Zuo, Yiping Zhang, Yi Sun.

**Visualization:** Fangbo Wen.

**Writing – original draft:** Jin Wang.

**Writing – review & editing:** Lulin Zheng, Wenjibin Sun.

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
