## [Decision Letter · Decision Letter 0]

8 Apr 2025

PONE-D-24-60368Study on deformation law of coal pore mechanism characteristics under peak cluster landformPLOS ONE

Dear Dr. Zheng,

Thank you for submitting your manuscript to PLOS ONE. After careful consideration, we feel that it has merit but does not fully meet PLOS ONE’s publication criteria as it currently stands. Therefore, we invite you to submit a revised version of the manuscript that addresses the points raised during the review process.

We look forward to receiving your revised manuscript.

Kind regards,

Zhengzheng Cao

Academic Editor

PLOS ONE

**Journal Requirements:**

1. When submitting your revision, we need you to address these additional requirements. Please ensure that your manuscript meets PLOS ONE's style requirements, including those for file naming. The PLOS ONE style templates can be found at https://journals.plos.org/plosone/s/file?id=wjVg/PLOSOne_formatting_sample_main_body.pdf and https://journals.plos.org/plosone/s/file?id=ba62/PLOSOne_formatting_sample_title_authors_affiliations.pdf 2. Thank you for stating in your Funding Statement: This research was supported by the Guizhou Province of Social Funding Project(LDLFJSFW2024-9). The funding had important roles in the study design, data collection and analysis.  Please provide an amended statement that declares *all* the funding or sources of support (whether external or internal to your organization) received during this study, as detailed online in our guide for authors at http://journals.plos.org/plosone/s/submit-now.  Please also include the statement “There was no additional external funding received for this study.” in your updated Funding Statement. Please include your amended Funding Statement within your cover letter. We will change the online submission form on your behalf. 3. We note that your Data Availability Statement is currently as follows: All relevant data are within the manuscript and its Supporting Information files. Please confirm at this time whether or not your submission contains all raw data required to replicate the results of your study. Authors must share the “minimal data set” for their submission. PLOS defines the minimal data set to consist of the data required to replicate all study findings reported in the article, as well as related metadata and methods (https://journals.plos.org/plosone/s/data-availability#loc-minimal-data-set-definition). For example, authors should submit the following data: - The values behind the means, standard deviations and other measures reported;- The values used to build graphs;- The points extracted from images for analysis. Authors do not need to submit their entire data set if only a portion of the data was used in the reported study. If your submission does not contain these data, please either upload them as Supporting Information files or deposit them to a stable, public repository and provide us with the relevant URLs, DOIs, or accession numbers. For a list of recommended repositories, please see https://journals.plos.org/plosone/s/recommended-repositories. If there are ethical or legal restrictions on sharing a de-identified data set, please explain them in detail (e.g., data contain potentially sensitive information, data are owned by a third-party organization, etc.) and who has imposed them (e.g., an ethics committee). Please also provide contact information for a data access committee, ethics committee, or other institutional body to which data requests may be sent. If data are owned by a third party, please indicate how others may request data access.

**Additional Editor Comments:**

You will see from the referees' comments that additional information needs to be provided, and we ask that this be provided, before we consider you manuscript further.

Reviewers' comments:

Reviewer's Responses to Questions

**Comments to the Author**

1. Is the manuscript technically sound, and do the data support the conclusions?

Reviewer #1: Yes

Reviewer #2: Yes

Reviewer #3: Yes

2. Has the statistical analysis been performed appropriately and rigorously? 

Reviewer #1: Yes

Reviewer #2: Yes

Reviewer #3: Yes

3. Have the authors made all data underlying the findings in their manuscript fully available?

Reviewer #1: Yes

Reviewer #2: Yes

Reviewer #3: Yes

4. Is the manuscript presented in an intelligible fashion and written in standard English?

Reviewer #1: Yes

Reviewer #2: Yes

Reviewer #3: Yes

5. Review Comments to the Author

**Reviewer #1: ** In this paper, aiming at the special peak cluster landform environment, the influence of vertical principal stress on the change law of coal rock pore structure under peak cluster landform is explored by using high pressure mercury injection and low temperature nitrogen adsorption experiment combined with fractal theory. The overall structure of the article is relatively perfect, but there are still some problems to be considered. The specific issues are as follows:

1. In Fig 1(a), the meaning is unclear, please explain the specific meaning of the figure.

2. There are many inconsistencies in the row spacing of the tables in the text, such as tables 4 and 5.

3. On pages 6, 8, and 14, the formula formats are inconsistent, please keep unified.

4. In lines 288 to 296, for the incomplete reference of the fractal theory method, appropriate references must be added to ensure academic rigor.

5. In line 400, the author uses Gaussian function to analyze the relationship between vertical principal stress and pore structure and fractal dimension. Please explain why Gaussian function is used instead of other functions and explain its necessity.

**Reviewer #2: ** The authors investigated the influence of vertical principal stresses on the change pattern of coal pore structure under the peak cluster landform by utilizing the experimental techniques of high-pressure mercury compression and low-temperature nitrogen adsorption in combination with the theoretical method of fractal dimension for the special peak cluster landform. The study was carried out systematically and the manuscript is clear in logic and complete in presentation, And this study presents interesting findings and contributes valuable insights into the study of coal pore structure under varying conditions.

However, the manuscript still has some shortcomings. Therefore, it is recommended that the manuscript be revised according to the following comments.

1. Page 4, what are the special conditions of the Peak Cluster landform? Please provide a detailed description in the text.

2. Page 6, formulas should be cited in the text and it should be ensured that the line height of formulas in the text remains consistent.

3. On pages 20 and 21, Figure and Fig are not consistent in the text, and abbreviations and acronyms are not consistent.

4. On pages 303-448, change the canonical terms such as fractal dimension D to italics in the text as requested.

5. On page 8, in the formula for calculating vertical principal stresses, is the value of stratum density taken into account in the field measurement data? It is recommended that the source of the density data be stated.

6. Page 14 and 15, pore systems can be described by a variety of fractal models. The authors have used the Frenkel-Halsey-Hill model and the Menger model to analyze pore structure complexity. Please explain the superiority of the two models in the description of pore structure complexity.

7. On page 18, this paper categorizes pores into adsorption and percolation pores, so please elaborate in the text on the reasons for analyzing only adsorption pores and bring in relevant literature to deepen the reader's understanding.

8.It is suggested that the author supplement the latest research literature on the theoretical methods of fractal dimension.

**Reviewer #3:**  The manuscript investigates the influence of vertical principal stress on the variation patterns of coal pore structure in peak cluster landforms, employing experimental techniques such as high-pressure mercury intrusion and low-temperature nitrogen adsorption, as well as the theoretical method of fractal dimension. The manuscript is logically structured and presents interesting perspectives on the characteristics of coal pore structure. However, minor revisions are required to address the following issues to ensure the completeness of the manuscript.

1. The article has limited discussion on the correlation between the peak cluster landform conditions and the vertical principal stress. It is recommended to strengthen the connection between the two at appropriate locations to enhance reader comprehension.

2. In Figure 7, there appears to be a missing data point for the LF4 coal sample, which results in an incomplete adsorption-desorption curve. It is advisable to conduct a thorough review and make the necessary corrections to ensure the integrity of the data presented.

3. The theories and conclusions regarding the fractal method presented in this article are incomplete. To enhance the academic rigor of the paper, it is essential to provide appropriate references that support the claims made.

4. In Section 4.1, the author categorizes pores into percolation pores and adsorption pores but lacks specific reasons for this distinction. It is suggested that the author add appropriate content and references to enhance reader comprehension.

5. In Section 4.2, the meanings and interpretations of the fractal dimensions D1 and D2 need further description to enhance reader understanding and readability.

6. The article employs a combination of mercury intrusion porosimetry and low-temperature nitrogen adsorption experiments. Please explain the advantages of combining these two methods for pore structure characterization and the necessity of establishing a critical pore diameter.

6. PLOS authors have the option to publish the peer review history of their article (what does this mean? ). If published, this will include your full peer review and any attached files.

**Do you want your identity to be public for this peer review?** For information about this choice, including consent withdrawal, please see our Privacy Policy .

Reviewer #1: No

Reviewer #2: No

Reviewer #3: No

---

## [Author Response · Author response to Decision Letter 1]

21 Apr 2025

Reviewer#1:

1. In Fig 1(a), the meaning is unclear, please explain the specific meaning of the figure.

Response and revise: thank you very much for your valuable feedback. In Fig. 1(a), the left figure shows the surface contour map of the study area, while the right figure shows the plan view of the underground 120910 working face of the study area based on the surface contour, and the upper and lower two lines are the 120910 return lane and 120910 transportation lane, so the coal samples of LF1-LF9 were taken from the 120910 working face based on the different contour lines. Detailed information can be found in section 2.1 of the revised manuscript.

2. There are many inconsistencies in the row spacing of the tables in the text, such as tables 4 and 5.

Response and revise: thank you very much for your valuable feedback. Based on your guidance, we have set the full text tables to be uniformly spaced.

3. On pages 6, 8, and 14, the formula formats are inconsistent, please keep unified.

Response and revise: thank you very much for your valuable feedback. Based on your guidance, we have set all formulas to a uniform size. The specific revisions are listed below:

1

2

3

4

5

6

4. In lines 288 to 296, for the incomplete reference of the fractal theory method, appropriate references must be added to ensure academic rigor.

Response and revise: thank you very much for your valuable feedback. Based on your guidance, the manuscript has added relevant references such as:

[42] Zuo T, Li XL, Wang JG, Hu QW, Tao ZH, Hu T. Insights into natural tuff as a building material: Effects of natural joints on fracture fractal characteristics and energy evolution of rocks under impact load. Eng. Fail. Anal., 2024, 108584. https://doi.org/10.1016/j.engfailanal.2024.108584.

[44] Wang L, Wu SW, Han SR, Hu B, Wang Q, Zhang KZ, Song T. Fractal Analysis of Coal Pore Structure Based on Low-Pressure Gas Adsorption and Its Influence on Methane Adsorption Capacity: A Perspective from Micropore Filling Model. Energy Fuels 2024 38(5), 4031-4046. https://doi.org/10.1021/acs.energyfuels.3c04724.

[46] Zhang JS, Ni XM, Liu XL, Su E. Influences of Different Acid Solutions on Pore Structures and Fractal Features of Coal. Fractal Fract. 2024, 8(2). https://doi.org/10.3390/fractalfract8020082.

[49] Pan HJ, Shao YC, Liu ZZ, Zuo QL, Su JT, Bai JL, Miao HY, Guo YQ, Zhang JJ. A Physical Parameter Characterizing Heterogeneity of Pore and Fracture Structures in Coal Reservoirs. Processes 2024, 12(8), 1553. https://doi.org/10.3390/pr12081553.

[50] Zhu HR, Chen Y. Wang HC, Ma DM, Li GF, Ji CJ, Wang X, Shi JB. Differences in pore characteristics of different macroscopic coal rock components in low rank coal based on multi scale joint characterization. J. Xi’an Univ. Sci. Technol., 2025, 45(2), 404-418. https://doi.org/10.13800/j.cnki.xakjdxxb.2025.0218.

[51] Liu JJ, Zhang YL, Yang D, Gao ZY, Wang D. Joint characterization of middle and low rank coal pores and fractures based on pressed mercury-nitrogen adsorption-nuclear magnetic resonance method. J. Henan Polytech. Univ. Nat. Sci., 2025, 44(2), 19-31. https://doi.org/10.16186/j.cnki.1673-9787.2024010059.

5. In line 400, the author uses Gaussian function to analyze the relationship between vertical principal stress and pore structure and fractal dimension. Please explain why Gaussian function is used instead of other functions and explain its necessity.

Response and revise: thank you very much for your valuable feedback. We believe that the use of Gaussian function can be well matched with the statistical characteristics of the data, and the Gaussian function can transform the complex stress-pore structure relationship into interpretable parameters through a concise mathematical form, which provides a highly efficient tool for quantitatively revealing the control law of the principal stresses on the physical properties of coal reservoirs. Other functions, either due to lack of flexibility or vague physical meaning, are difficult to replace the analytical effect of Gaussian function under the same conditions. For example, although high-order polynomials can flexibly fit complex curves, they easily lead to overfitting phenomena and the physical meaning of the parameters is unclear, while the exponential function is only applicable to monotonically increasing or decreasing processes, and cannot describe the peak phenomena presented in the experimental results. Therefore, Gaussian function is used in this study to analyze the relationship between vertical principal stress and pore structure and fractal dimension.

Based on your guidance, we have added the description of choosing Gaussian function in section 4.1 of the revised manuscript. The specific descriptions are listed below:

According to the experimental results, the vertical principal stresses show a relatively symmetric peak phenomenon with respect to the pore structure and fractal dimension, which cannot be reflected by basic functions such as polynomial and exponential functions, for example, while the Gaussian function can naturally fit this statistical property, and its bell curve can effectively describe the concentration trend and the degree of discretization of the variables around the peak value.

Reviewer #2:

1. Page 4, what are the special conditions of the Peak Cluster landform? Please provide a detailed description in the text.

Response and revise: thank you very much for your valuable feedback.As a key component of the karst landform system, the peaked landforms are known for their outstanding uniqueness and structural complexity. The formation of these landforms is mainly attributed to the long-term influence of karst, and is characterized by a unique landscape consisting of isolated peaks and pillars with interspersed valleys and basins. In Guizhou, the vertical difference of the peaked landforms is especially significant, forming a typical steep topography, and the formation mechanism of this vertical height difference is closely related to the karst action and the regional geological structure. Under the influence of continuous water erosion, the difference in elevation between the peaks and gullies can be hundreds or even thousands of meters, resulting in high variability in the overlying rock stress.

Based on your guidance, we have added descriptions to the paper. Detailed information can be found in the introduction of the revised manuscript. The specific descriptions are listed below:

But the geological conditions of coal reservoirs, as an important factor restricting the development of coalbed methane, compared with other karst landforms, the peak cluster landforms between the peaks of the peak district is highly undulating., with a vertical height difference of hundreds of meters and steep slopes, resulting in high variability of the rock stresses overlying the coal reservoirs, which not only influences the complexity of the coal's pore structure, but also controls the gas transport channels of the coal reservoirs.

2. Page 6, formulas should be cited in the text and it should be ensured that the line height of formulas in the text remains consistent.

Response and revise: thank you very much for your valuable feedback. Based on your guidance, we have set all formulas to a uniform line height and added references to Eq. (1), (4), (5) and (6) in the text. The specific descriptions are listed below:

According to Eq. (1) Washburn's formula.

1

4

Based on Eq. (4) the data related to ln(V/Vm) and ln[ln(P0/P)] are fitted, as shown in Fig 9, the slope of the fitted line segment in the graph is the fractal dimension value.

Menger model for mesopore and macropore fractal dimension D is calculated as shown in Eq.(5).

5

6

Based on Eq. (6), the vertical principal stress is fitted to the adsorption pore SSA and pore volume, and the fitting plot is shown in Fig 12.

3. On pages 20 and 21, Figure and Fig are not consistent in the text, and abbreviations and acronyms are not consistent.

Response and revise: thank you very much for your valuable feedback. Based on your guidance, we have harmonized the Figure with the Fig representation on pages 20 and 21.

4. On pages 303-448, change the canonical terms such as fractal dimension D to italics in the text as requested.

Response and revise: thank you very much for your valuable feedback. Based on your guidance, we have changed all relevant canonical terms covered in the text to italics.

5. On page 8, in the formula for calculating vertical principal stresses, is the value of stratum density taken into account in the field measurement data? It is recommended that the source of the density data be stated.

Response and revise: thank you very much for your valuable feedback. The stratum density of coal rock seams involved in this paper is taken based on the basic geological characteristics of the study area, so the stratum density value in the vertical principal stress calculation formula takes into account the geological borehole data and field measurement data of the study area. Based on your guidance, we have added relevant descriptions in the paper. Detailed information can be found in section 3.1 of the revised manuscript. The specific revisions are listed below:

Based on the geological borehole data and field measurements collected in the study area, the rock density for each layer of coal rock has been recorded, in which the average density of sandstone is 2500kg/m3, the average density of limestone is 2650kg/m3, the rock density of mudstone is 2300kg/m3, and the average density of coal is 1300kg/m3.

6. Page 14 and 15, pore systems can be described by a variety of fractal models. The authors have used the Frenkel-Halsey-Hill model and the Menger model to analyze pore structure complexity. Please explain the superiority of the two models in the description of pore structure complexity.

Response and revise: thank you very much for your valuable feedback. The Frenkel-Halsey-Hill model is based on the multilayer adsorption theory and quantifies the fractal characteristics of the pore surfaces through adsorption isotherm data.The FHH model calculates the fractal dimension (D) through the relationship between the relative pressure in the adsorption isotherm and the amount of adsorption. As D approaches 2, the surface is smooth; as D approaches 3, the pores exhibit a highly irregular layered structure, and this correlation provides a direct tool for quantifying the geometric complexity of the pore surface. In addition, the Frenkel-Halsey-Hill model not only reflects the surface fractal but also indirectly reveals the pore connectivity by analyzing the differences in the adsorption-desorption hysteresis loop. And the value of fractal dimension D obtained by the Menger model is directly related to the geometric complexity of pore network such as pore inhomogeneity, and can explain the reason for the in-and-out Hg hysteresis in the Hg compression curve. When D is close to 2, adsorption within the pore space is mainly dominated by open fractures; when D is close to 3, volume filling of isolated pores with poor connectivity within the pore space is a significant feature, and this correlation can intuitively explain the fractal behaviors of the different pressure segments in the piezomercury curves. Detailed information can be found in section 3.3 of the revised manuscript.

7. On page 18, this paper categorizes pores into adsorption and percolation pores, so please elaborate in the text on the reasons for analyzing only adsorption pores and bring in relevant literature to deepen the reader's understanding.

Response and revise: thank you very much for your valuable feedback. It is generally believed that pore size greater than 100nm is seepage pore and less than 100nm is adsorption pore. Coal gas adsorption is dominated by physical adsorption, which mainly occurs in micropore and transition pore with a pore diameter of less than 100nm, which have a very high specific surface area in the pores, accounting for more than 80% of the total specific surface area, and provide a large number of adsorption sites for gas molecules, which directly determines the amount of adsorption of coal gas. The mesopore and macropore with a pore size larger than 100 nm are mainly used as a channel for gas transportation, with a smaller specific surface area and a weaker contribution to the adsorption amount. In addition the gas in the seepage pores mostly exists in the free state rather than in the adsorbed state, so in the study of the effect of pore structure on gas adsorption, it is often considered only the adsorption pores, but not the seepage pores. Based on your guidance, we have added relevant descriptions in the paper. Detailed information can be found in section 4.1 of the revised manuscript. The specific revisions are listed below:

Based on previous studies, this paper defines pore size greater than 100nm as seepage pore and pore size less than 100nm as adsorption pore[52]. Coal gas adsorption is dominated by physical adsorption, which occurs on the coal surface, and gas molecules are more easily adsorbed on the pore surface when the pores have a high specific surface area. In addition, the gas mainly permeates in the form of laminar or turbulent flow in the macropore and mesopore , and exists in the form of capillary condensate or physical adsorption film on the pore wall in the micropore and transition pore, which indicates that the seepage pore mainly serves as a gas transport channel and the seepage pore has a small specific surface area, which makes a limited contribution to the amount of adsorption, while the adsorption pore has a very high specific surface area in the pore, accounting for more than 80% of the total specific surface area, which provides a large number of adsorption sites for the gas molecules provide a large number of adsorption sites, which directly determines the amount of coal gas adsorption[53]. Therefore, in order to study the differences in the pore structure of coal under the influence of vertical principal stress, this section mainly focuses on the variation rules of specific surface area and pore volume of adsorption pores with vertical principal stress.

[52] Li ZW, Lin BQ, Hao ZY, Gao YB. Characteristics ofpore size distribution of coal and its impacts on gas adsorption. J. China Univ. Min. Technol., 2013, 42(6): 1047-1053. https://doi.org/10.13247/j.cnki.jcumt.2013.06.025.

[53] Yang H, Bi WY, Zhang YG, Yu JK, Yan JW, Lei DJ, Ma ZN. Effect of tectonic coal structure on methane adsorption, J. Environ. Chem. Eng., 2021, 9(6), 106294. https://doi.org/10.1016/j.jece.2021.106294.

8.It is suggested that the author supplement the latest research literature on the theoretical methods of fractal dimension.

Response and revise: thank you very much for your valuable feedback. Based on your guidance, the manuscript has added relevant references such as:

[42] Zuo T, Li XL, Wang JG, Hu QW, Tao ZH, Hu T. Insights into natural tuff as a building material: Effects of natural joints on fracture fractal characteristics and energy evolution of rocks under impact load. Eng. Fail. Anal., 2024, 108584. https://doi.org/10.1016/j.engfailanal.2024.108584.

[44] Wang L, Wu SW, Han SR, Hu B, Wang Q, Zhang KZ, Song T. Fractal Analysis of Coal Pore Structure Based on Low-Pressure Gas Adsorption and Its Influence on Methane Adsorption Capacity: A Perspective from Micropore Filling Model. Energy Fuels 2024 38(5), 4031-4046. https://doi.org/10.1021/acs.energyfuels.3c04724.

[46] Zhang JS, Ni XM, Liu XL, Su E. Influences of Different Acid Solutions on Pore Structures and Fractal Features of Coal. Fractal Fract. 2024, 8(2). https://doi.org/10.3390/fractalfract8020082.

[49] Pan HJ, Shao YC, Liu ZZ, Zuo QL, Su JT, Bai JL, Miao HY, Guo YQ, Zhang JJ. A Physical Parameter Characterizing Heterogeneity of Pore and Fracture Structures in Coal Reservoirs. Processes 2024, 12(8), 1553. https://doi.org/10.3390/pr12081553.

[50] Zhu HR, Chen Y. Wang HC, Ma DM, Li GF, Ji CJ, Wang X, Shi JB. Differences in pore characteristics of different macroscopic coal rock components in low rank coal based on multi scale joint characterization. J. Xi’an Univ. Sci. Technol., 2025, 45(2), 404-418. https://doi.org/10.13800/j.cnki.

---

## [Decision Letter · Decision Letter 1]

13 May 2025

PONE-D-24-60368R1Study on deformation law of coal pore mechanism characteristics under peak cluster landformPLOS ONE

Dear Dr. Zheng,

Thank you for submitting your manuscript to PLOS ONE. After careful consideration, we feel that it has merit but does not fully meet PLOS ONE’s publication criteria as it currently stands. Therefore, we invite you to submit a revised version of the manuscript that addresses the points raised during the review process.

We look forward to receiving your revised manuscript.

Kind regards,

Zhengzheng Cao

Academic Editor

PLOS ONE

Reviewers' comments:

Reviewer's Responses to Questions

**Comments to the Author**

1. If the authors have adequately addressed your comments raised in a previous round of review and you feel that this manuscript is now acceptable for publication, you may indicate that here to bypass the “Comments to the Author” section, enter your conflict of interest statement in the “Confidential to Editor” section, and submit your "Accept" recommendation.

Reviewer #4: All comments have been addressed

Reviewer #5: (No Response)

Reviewer #6: (No Response)

Reviewer #7: All comments have been addressed

2. Is the manuscript technically sound, and do the data support the conclusions?

Reviewer #4: Partly

Reviewer #5: Yes

Reviewer #6: (No Response)

Reviewer #7: Yes

3. Has the statistical analysis been performed appropriately and rigorously? 

Reviewer #4: Yes

Reviewer #5: Yes

Reviewer #6: (No Response)

Reviewer #7: Yes

4. Have the authors made all data underlying the findings in their manuscript fully available?

Reviewer #4: Yes

Reviewer #5: Yes

Reviewer #6: (No Response)

Reviewer #7: Yes

5. Is the manuscript presented in an intelligible fashion and written in standard English?

Reviewer #4: Yes

Reviewer #5: Yes

Reviewer #6: (No Response)

Reviewer #7: Yes

6. Review Comments to the Author

Reviewer #4: This paper conducts pore characteristic analysis on 9 sample. A case about the variation in coal pore structure beneath peak cluster topography. The paper provides some information and data on the region, which can serve as a reference for readers or researchers. The authors have answered all the comments of three anonymous review experts. The biggest problem with this paper is that it did not provide any new insights, as there have been many studies on the pores and fractures of coal, and conventional methods were used in this paper. For data analysis, fractals were used. Suggest adding previous literature data in the sub parameter section for thorough discussion.

According to reviewer 3, Please explain the advantages of combining these two methods for pore structure characterization and the necessity of establishing a critical pore diameter. This comment is very important, and it is recommended that the author further revise it.

Reviewer #5: After the previous round of revisions, the quality of the manuscript has been significantly improved. I believe that the manuscript has basically met the requirements for publication. But there are still a few issues that need further modification. The following comments and suggestions should guide the authors to revise the paper. 1�It is suggested that the authors further supplement the application significance and value of the research after the discussion section. This will be of great reference significance for researchers and engineers in related fields. 2�I noticed that some previous research literature related to this study was not cited in the article, such as previous research cases on coal pore structure and fractal characteristics. It is suggested that the authors further supplement the relevant literature. 3�The value and significance of new insights and findings in the manuscript are not highlighted in the abstract and conclusion.

Reviewer #6: 1. The article is comprehensive in experimental design, covering coal sample analysis under different geological conditions and stress states. However, in order to enhance the reliability of the experiment and the statistical significance of the data, it is recommended to further discuss the possible sources of error in the experiment and consider increasing the number of repeated experiments. In addition, a more detailed error analysis should be provided for the processing and analysis of the experimental data to ensure the robustness of the conclusions.

2. The method proposed in the article combines high-pressure mercury injection and low-temperature nitrogen adsorption experiments, and combines fractal theory to perform multi-scale quantitative characterization of the pore structure of coal samples, which shows significant innovation. It is recommended to further clarify the advantages of this method over traditional single characterization techniques and discuss in depth its potential application in the study of pore structure of other porous media (such as rocks, soils, etc.).

3. Although the article compares and analyzes the experimental results and numerical simulation results in detail, there are some inconsistent results. For example, the simulated permeability of some coal samples is higher than the experimental value. It is recommended to conduct a more in-depth analysis of these inconsistencies, explore possible sources of simulation errors (such as model simplification, parameter setting, etc.), and propose corresponding improvement measures to improve the accuracy of the simulation.

4. The article simplifies the formation of contact bridges when using DEM to simulate the compaction process of coal samples, which may affect the accuracy of the simulation results. It is recommended to further explore the specific impact of this simplification on the complexity of pore structure and permeability simulation results, and consider using more sophisticated geometric modeling methods (such as considering the irregularity of particle shape, dynamic changes in contact force, etc.) to improve the authenticity and accuracy of the simulation.

5. The abstract of the article needs to be revised, and the importance of the article should be highlighted. There are insufficient references, so more references need to be supplemented. There are too few references, which need to be supplemented to 30. The background and mechanism are not introduced clearly.

Mechanical behavior and fracture mechanism of high-temperature granite cooled with liquid nitrogen for geothermal reservoir applications. Physics of Fluids 2025; 37 (2): 026616. https://doi.org/10.1063/5.0253668

6. When comparing the experimental and simulation data, the article mainly shows the average value, but does not provide statistics such as standard deviation and confidence interval. In order to more comprehensively evaluate the discreteness and consistency of the data, it is recommended to supplement these statistics. In addition, for the systematic deviation that may occur in the simulation, a sensitivity analysis should be conducted to determine the source of the deviation (such as model parameters, boundary conditions, etc.), and an effective correction method should be proposed to improve the reliability and practicality of the simulation results.

Reviewer #7: 1. The annotations on some charts (as shown in Figures 3 and 4) are not clear enough, especially regarding the units of the coordinate axes and the explanations of the legends. It is suggested to optimize the chart design and enhance the clarity of the chart to ensure that readers can understand the data intuitively.

2. Only 9 coal samples were collected in the text, which may be insufficient to cover different geological conditions under peak cluster landforms. It is suggested that the basis for sample selection be elaborated in detail in the methods section.

3. Some research work can be useful in the introduction. Study on the degradation mechanism of mechanical properties of red sandstone under static and dynamic loading after different high temperatures. 

4. In the data analysis section, using the Gaussian function to fit the data is innovative, but it only presents the fitting results and coefficients, lacking a detailed discussion of the goodness of fit test results.

7. PLOS authors have the option to publish the peer review history of their article (what does this mean? ). If published, this will include your full peer review and any attached files.

**Do you want your identity to be public for this peer review?** For information about this choice, including consent withdrawal, please see our Privacy Policy .

Reviewer #4: No

Reviewer #5: No

Reviewer #6: No

Reviewer #7: No

---

## [Author Response · Author response to Decision Letter 2]

26 Jun 2025

Reviewer #4:

This paper conducts pore characteristic analysis on 9 sample. A case about the variation in coal pore structure beneath peak cluster topography. The paper provides some information and data on the region, which can serve as a reference for readers or researchers. The authors have answered all the comments of three anonymous review experts. The biggest problem with this paper is that it did not provide any new insights, as there have been many studies on the pores and fractures of coal, and conventional methods were used in this paper. For data analysis, fractals were used. Suggest adding previous literature data in the sub parameter section for thorough discussion.

Response and revise: thank you very much for your valuable feedback. We fully understand and partially agree with your perspective. As you have pointed out, there is already a substantial body of literature in the field of research on coal pore and fracture structures, and this paper indeed employs analytical methods that are well-established and conventional in this field. However, we would like to emphasize the core objective and research context of this study. Our research is not aimed at proposing new theories or groundbreaking methods, but rather focuses on the specific geological context of peak cluster landform. We systematically characterize these landform using mature and reliable experimental methods such as high-pressure mercury intrusion and low-temperature nitrogen adsorption. Our primary contribution lies in providing detailed pore structure characteristics under the conditions of peak cluster landform, combined with fractal dimensions to gain a more comprehensive understanding. To more clearly highlight the value and positioning of this research, we have further elaborated on the uniqueness of the study area and the practical needs and application background of studying the pore structure in this region in the discussion section of the paper. The specific descriptions are listed below:

The geological conditions of coal reservoirs are an important factor in the development of CBM, and as a key component of the karst geomorphology system, the coal under this geomorphology has obvious vertical variability, which makes the pore structure of the coal more complex, and leads to the difficulty in clarifying the law of gas enrichment under this geomorphology. Therefore, the study of the multi-scale pore structure characteristics of coal and its complexity under the peak cluster landform can further provide reference for the study of the gas storage and enrichment law, which can not only deepen the gas storage law under special geological conditions, but also lay the foundation for the key technologies of resource development, disaster prevention and control in the similar topography, so as to guarantee the safety of mine mining.

According to reviewer 3, Please explain the advantages of combining these two methods for pore structure characterization and the necessity of establishing a critical pore diameter. This comment is very important, and it is recommended that the author further revise it.

Response and revise: thank you very much for your valuable feedback. It was previously mentioned in response to Reviewer 3's question that the mercury pressing experiment is based on the Washburn equation, which infiltrates mercury into pores by applying pressure, and it is sensitive to mesopors, macropore, and fissures, etc., and is especially suitable for characterizing open-type connected pores and seepage channels, but it is limited by the surface tension and the contact angle of mercury, which leads to the extremely low efficiency of mercury intrusion into micro- and small pores; whereas, the low-temperature nitrogen adsorption experiments are based on the BET model and the BJH model, through the adsorption-desorption isotherm of nitrogen at low temperature, it can accurately analyze the micro- and small pores, but the sensitivity of characterization of medium- and large-sized pores is low. Therefore, the combination of Hg-pressing experiments and low-temperature nitrogen adsorption realizes the full-scale coverage of pores and makes up for the pore size detection blind spot of a single method. The establishment of critical pore size is to reduce the error of the experimental results, because the pore structure of coal body analyzed by mercury pressure experiment and low temperature nitrogen gas adsorption experiment will lead to part of the data overlap, resulting in a large error in the experimental data, so the determination of the critical pore size can guarantee the accuracy of the analysis of the pore structure of the combination of the mercury pressure experiment and the low-temperature nitrogen gas adsorption experiment.

Based on your guidance, we have added descriptions to the paper. Detailed information can be found in section 3.2.3 of the revised manuscript. The specific descriptions are listed below:

Since the mercury pressure experiment is based on the pressure of mercury liquid to measure the pore size, at the beginning of the experiment, the mercury liquid was first pressed into the cracks of the coal samples, and then gradually pressed into the pores with the increase of pressure. However, as the pressure increases, some pores cannot resist the pressure of mercury liquid to form a large number of fissures, resulting in experimental deviation, and limited by the surface tension and contact angle of mercury, resulting in extremely low efficiency of mercury intrusion into micro and small pores; low-temperature liquid nitrogen adsorption is based on the cohesion of liquid nitrogen to fill the pores to detect the pore size, and is able to accurately analyze the micro and small pores, but has low sensitivity to the characterization of the medium and large pores, mainly because when measuring large pore sizes, the liquid nitrogen cannot coalesce.

The main reason is that when measuring the large pore size, the liquid nitrogen can not be coalesced with the increasing pressure, which makes the large pore size data of the measured coal samples have a large deviation. Therefore, characterizing the pore structure of coal by combining the two experiments not only realizes the full-scale coverage of pores and makes up for the blind spot of pore diameter detection of a single method, but also provides a more comprehensive and reliable data basis for the characterization of the pore structure of coal through the synergistic analysis of the morphology, specific surface area, and fractal features obtained from the two experiments. In addition, because the pore structure of coal can be partially overlapped when analyzing the pore structure of mercury pressure and low-temperature nitrogen adsorption experiments, resulting in a large error in the experimental data, based on the different sensitivities of the results of the two experiments for the characterization of multi-scale pores, a critical pore diameter was established as a pore diameter indicator to distinguish between the results of mercury pressure experiments and low-temperature nitrogen adsorption experiments, and the critical pore diameter was determined as the intersection point rc of the change curves of the measured pore diameters and the pore volumes of the two experiments.

The critical pore size was determined mainly as the intersection of two experimentally determined pore size versus pore volume curves rc. If the diameter of the pore is less than rc, data from the adsorption experiment with liquid nitrogen at low temperatures is utilized. Conversely, when the diameter of the pore is larger than rc, the data from the mercury intrusion experiment is utilized. Since some coal samples exhibit multiple intersections of the curves derived from both experiments, the last intersection point is selected as the critical pore diameter rc for that particular coal sample. The determination of the critical pore size guarantees the accuracy of the pore structure analysis in the combination of mercury pressure experiments and low-temperature nitrogen adsorption experiments.

Reviewer #5:

After the previous round of revisions, the quality of the manuscript has been significantly improved. I believe that the manuscript has basically met the requirements for publication. But there are still a few issues that need further modification. The following comments and suggestions should guide the authors to revise the paper.

1.It is suggested that the authors further supplement the application significance and value of the research after the discussion section. This will be of great reference significance for researchers and engineers in related fields.

Response and revise: thank you very much for your valuable feedback. Based on your guidance, we have added descriptions to the paper. Detailed information can be found in the discussion of the revision. The specific descriptions are listed below:

The geological conditions of coal reservoirs are an important factor in the development of CBM, and as a key component of the karst geomorphology system, the coal under this geomorphology has obvious vertical variability, which makes the pore structure of the coal more complex, and leads to the difficulty in clarifying the law of gas enrichment under this geomorphology. Therefore, the study of the multi-scale pore structure characteristics of coal and its complexity under the peak cluster landform can further provide reference for the study of the gas storage and enrichment law, which can not only deepen the gas storage law under special geological conditions, but also lay the foundation for the key technologies of resource development, disaster prevention and control in the similar topography, so as to guarantee the safety of mine mining.

2.I noticed that some previous research literature related to this study was not cited in the article, such as previous research cases on coal pore structure and fractal characteristics. It is suggested that the authors further supplement the relevant literature.

Response and revise: thank you very much for your valuable feedback. Based on your guidance, the manuscript has added relevant references such as:

[30] Zhao DF, Zhang JM, Guan X, Liu DD, Wang QX, Jiao WW, Zhou XQ, Li YJ, Wang G, Guo YH. Comparing the Pore Networks of Coal, Shale, and Tight Sandstone Reservoirs of Shanxi Formation, Qinshui Basin: Inspirations for Multi-Superimposed Gas Systems in Coal-Bearing Strata. Appl. Sci. 2023; 13(7):4414. https://doi.org/10.3390/app13074414.

[44] Zhao DF, Guo YH, Wang G, Guan X, Zhou XQ, Liu J. Fractal Analysis and Classification of Pore Structures of High-Rank Coal in Qinshui Basin, China[J]. Energies. 2022; 15(18):6766. https://doi.org/10.3390/en15186766.

[58] Zhao DF, Guo YH, Wang G, Mao XX. Characterizing nanoscale pores and its structure in coal: Experimental investigation. Energ Explor Exploit, 2019, 37(4):1-28. https://doi.org/10.1177/0144598719831397.

3.The value and significance of new insights and findings in the manuscript are not highlighted in the abstract and conclusion.

Response and revise: thank you very much for your valuable feedback. Based on your guidance, we have added descriptions to the paper. Detailed information can be found in the Abstract and Conclusion of the revision. The specific descriptions are listed below:

Abstract:

In order to reveal the change rule of coal pore structure under the peak cluster landform, coal samples were taken from nine different mountain heights based on the vertical variability of the landform, and the pore structure of the coal samples was tested using a combination of high-pressure mercuric pressure method and low-temperature nitrogen adsorption experiments. The results show that compared with the traditional coal reservoir, the pore structure of coal under the peaked bush landscape, such as pore content, specific surface area and pore volume, changes with the change of vertical principal stress in a multi-peak state. The variations in the maximum and minimum values of vertical principal stress at each peak level are 1.04, 1.04, and 1.05 times, respectively. In terms of the adsorption pore volume, the differences between the maximum and minimum values are 2.30, 1.60, and 1.53 times, respectively. Notably, the degree of change between the peaks decreases as the peaks progress. Furthermore, with the increase in vertical principal stress, the degree of change in the specific surface area and pore volume of the corresponding adsorption pore between peaks also diminishes. It shows that the role of peak cluster landform conditions on coal pore structure is significant, and the extent of the role decreases with the increase of vertical principal stresses. Additionally, the vertical principal stress predominantly influences the fractal dimension D1, which is represented as pore surface roughness. The capacity of coal samples for gas adsorption is mainly influenced by the roughness of the pore surfaces and the volume of the adsorption pores. In summary, the degree of microcrack formation in the pores of coal samples is influenced to some extent by the vertical elevation difference characteristics of the peak cluster landform, which not only controls the characteristics of the pore structure, but also affects the gas adsorption capacity of the coal samples. These results highlight the influence of vertical principal stress on coal pore closure and structural changes under the peak cluster landform. The results of the study can provide a reference for further research on further gas storage and enrichment laws, and the mine can judge the risk of protrusion for the gas accumulation capacity of coal under the peaked cluster landform, so as to formulate effective gas prevention and control measures in advance.

Conclusion:

In this paper, the combination of high-pressure mercury pressure and low-temperature nitrogen adsorption testing method was used to quantitatively and comprehensively characterize the change rule of coal microporous structure under the influence of vertical height difference characteristics of the peak cluster landform, and based on the fractal theory, the fractal dimensions of the seepage pores and adsorption pores of the coal were respectively calculated, and then the change of the coal pore structure characteristics and its complexity under the peak cluster landform was investigated. The following conclusions were obtained:

(1)Influenced by the vertical elevation difference characteristics of the peak cluster landform, the coal is subjected to a gradual increase in vertical principal stresses in the overlying rock strata as the elevation of the mountain increases. Unlike individual techniques, the combined use of mercury injection and low-temperature nitrogen adsorption experiments provides better accuracy and thoroughness in analyzing the multi-scale pore structure of coal. The pore content obtained from this combined approach shows clear multi-peak variations that correspond to fluctuations in vertical principal stress.

(3) Compared with conventional coal reservoirs, the significant vertical height difference of the peak cluster landform leads to an obvious gradient of vertical principal stress, which exacerbates the non-homogeneous differentiation of pore structure. The fractal dimensions of adsorption pores, D1 and D2, show multi-peak variations with vertical principal stress, and the pore surface roughness of adsorption pores of coal samples is controlled by the vertical principal stress gradient under the peak cluster landform to a certain extent. In summary, the degree of closure of the pores of the coal samples under the crested clump topography varies in complexity with the vertical principal stress, which not only affects the pore structure characteristics of the coal samples, but also influences the gas adsorption capacity of the coal samples.

Reviewer #6:

1. The article is comprehensive in experimental design, covering coal sample analysis under different geological conditions and stress states. However, in order to enhance the reliability of the experiment and the statistical significance of the data, it is recommended to further discuss the possible sources of error in the experiment and consider increasing the number

---

## [Decision Letter · Decision Letter 2]

1 Aug 2025

Study on deformation law of coal pore mechanism characteristics under peak cluster landform

PONE-D-24-60368R2

Dear Dr. Zheng

We’re pleased to inform you that your manuscript has been judged scientifically suitable for publication and will be formally accepted for publication once it meets all outstanding technical requirements.

Kind regards,

Veer Singh, Ph.D

Academic Editor

PLOS ONE

Additional Editor Comments (optional):

Dear Author,

I would like to inform you that your manuscript is suitable for publication in PLOS One. 

Reviewers' comments:

Reviewer's Responses to Questions

**Comments to the Author**

1. If the authors have adequately addressed your comments raised in a previous round of review and you feel that this manuscript is now acceptable for publication, you may indicate that here to bypass the “Comments to the Author” section, enter your conflict of interest statement in the “Confidential to Editor” section, and submit your "Accept" recommendation.

Reviewer #4: All comments have been addressed

Reviewer #5: (No Response)

Reviewer #7: All comments have been addressed

2. Is the manuscript technically sound, and do the data support the conclusions?

Reviewer #4: Yes

Reviewer #5: Yes

Reviewer #7: Yes

3. Has the statistical analysis been performed appropriately and rigorously? 

Reviewer #4: Yes

Reviewer #5: Yes

Reviewer #7: Yes

4. Have the authors made all data underlying the findings in their manuscript fully available?

Reviewer #4: Yes

Reviewer #5: Yes

Reviewer #7: Yes

5. Is the manuscript presented in an intelligible fashion and written in standard English?

Reviewer #4: Yes

Reviewer #5: Yes

Reviewer #7: Yes

6. Review Comments to the Author

Reviewer #4: The author has made revisions and improvements according to the opinions of all experts. Although there are no new findings in the paper, the relevant data can be used as a reference for readers.

Reviewer #5: After revision, the quality of the manuscript has been further improved. The author information for reference [11] is missing. It is suggested that the authors supplement it.

Reviewer #7: accept

In order to reveal the change rule of coal pore structure under the peak cluster landform, coal samples were taken from nine different mountain heights based on the vertical variability of the landform, and the pore structure of the coal samples was tested using a combination of high-pressure mercuric pressure method and low-temperature nitrogen adsorption experiments. The results show that compared with the traditional coal reservoir, the pore structure of coal under the peak cluster landform, such as pore content, specific surface area and pore volume, changes with the change of vertical principal stress in a multi-peak state.

7. PLOS authors have the option to publish the peer review history of their article (what does this mean? ). If published, this will include your full peer review and any attached files.

**Do you want your identity to be public for this peer review?** For information about this choice, including consent withdrawal, please see our Privacy Policy .

Reviewer #4: No

Reviewer #5: No

Reviewer #7: No

---

## [Editor Report · Acceptance letter]

PONE-D-24-60368R2

PLOS ONE

Dear Dr. Zheng,

I'm pleased to inform you that your manuscript has been deemed suitable for publication in PLOS ONE. Congratulations! Your manuscript is now being handed over to our production team.

Kind regards,

on behalf of

Dr. Veer Singh

Academic Editor

PLOS ONE